# Maturation of cortical input to dorsal raphe nucleus increases behavioral persistence in mice

**Nicolas Gutierrez-Castellanos[1†], Dario Sarra[1,2†‡], Beatriz S Godinho[1,2‡], Zachary F Mainen[1]\***

[1]Champalimaud Research, Champalimaud Foundation, Lisbon, Portugal; [2]Nuffield Department of Clinical Neurosciences, University of Oxford, Oxford, United Kingdom

**Abstract** The ability to persist toward a desired objective is a fundamental aspect of behavioral control whose impairment is implicated in several behavioral disorders. One of the prominent features of behavioral persistence is that its maturation occurs relatively late in development. This is presumed to echo the developmental time course of a corresponding circuit within late-maturing parts of the brain, such as the prefrontal cortex, but the specific identity of the responsible circuits is unknown. Here, we used a genetic approach to describe the maturation of the projection from layer 5 neurons of the neocortex to the dorsal raphe nucleus in mice. Using optogenetic-assisted circuit mapping, we show that this projection undergoes a dramatic increase in synaptic potency between postnatal weeks 3 and 8, corresponding to the transition from juvenile to adult. We then show that this period corresponds to an increase in the behavioral persistence that mice exhibit in a foraging task. Finally, we used a genetic targeting strategy that primarily affected neurons in the medial prefrontal cortex, to selectively ablate this pathway in adulthood and show that mice revert to a behavioral phenotype similar to juveniles. These results suggest that frontal cortical to dorsal raphe input is a critical anatomical and functional substrate of the development and manifestation of behavioral persistence.

**\*For correspondence:** zmainen@neuro.fchampalimaud.org

[†]These authors contributed equally to this work

**Present address:** [‡]Nuffield Department of Clinical Neurosciences, University of Oxford, Oxford, United Kingdom

**Competing interest:** The authors declare that no competing interests exist.

## Editor's evaluation

In this important study, the authors explore the importance of developmental changes in cortico-DRN innervation in the balance of behavioral control in a foraging task. The authors report somewhat convincing evidence that while juvenile mice and adult mice both perform the task, juveniles exhibit more impulsive behavior due to reduced efficacy of cortico-DRN projections. The authors conclude that the development of cortico-DRN projections allows 5HT input to promote perseveration (or exploitation) in the balance of behavioral control.

## Introduction

Multiple aspects of behavioral control, including attention, cognitive flexibility, and behavioral persistence, emerge during critical periods of postnatal development. In these periods, environment and experience contribute to the maturation of higher cognitive functions (*Larsen and Luna, 2018*; *Mischel et al., 1989*; *Tooley et al., 2021*), which sets the foundations of future social and cognitive abilities during adulthood (*Casey et al., 2011*; *Moffitt et al., 2011*).

Ethologically, the development of behavioral control is critical for selective fitness and, thus, survival. For instance, in the natural environment, food resources are often sparsely distributed and depleted with consumption. Therefore, the well-known tradeoff between exploiting a depleting

resource and exploring in search of alternatives is crucial to reach an optimal foraging strategy and obtain the maximum amount of resources with minimal waste of physical effort. Therefore, a forager in a possibly depleted patch of food faces an important dilemma—to stay and continue to forage at the site or to leave and travel to another—that calls for a careful balancing between persistence (staying) and flexibility (leaving) (*Charnov, 1976*; *Lottem et al., 2018*; *Morris and Davidson, 2000*; *Vertechi et al., 2020*).

From a neural perspective, cognitive development correlates with large-scale synaptic and structural changes (*Durston and Casey, 2006*; *Shaw et al., 2006*; *Zuo et al., 2010*; *Zuo et al., 2017*) that are considered to underlie the emergence of increasing cognitive control over innate impulsive behavioral tendencies (*Alexander-Bloch et al., 2013*; *Fair et al., 2009*; *Luna et al., 2001*). A variety of evidence links the medial prefrontal cortex (mPFC) to the expression of behavioral control in a wide range of mammal species. For instance, humans and macaques with prefrontal cortical damage display deficits in behavioral flexibility, decision making, and emotional processing (*Izquierdo et al., 2017*; *Rudebeck et al., 2013*; *Roberts et al., 1998*), as well as a notable increase in impulsive behavior (*Berlin et al., 2004*; *Dalley and Robbins, 2017*; *Fellows, 2006*; *Itami and Uno, 2002*), all of which, at least partially, recapitulate features of juvenile behavior over healthy development in humans, non-human primates, and rodents (*Rosati et al., 2023*; *Doremus-Fitzwater et al., 2012*; *Romer, 2010*; *Weed et al., 2008*). In line with this, local pharmacological inhibition of mPFC significantly limits rats' ability to wait for a delayed reward (*Murakami et al., 2017*; *Narayanan et al., 2006*).

Crucially, the mPFC undergoes intense postnatal maturation from childhood to adulthood, particularly during adolescence (*Chini and Hanganu-Opatz, 2021*), which in humans spans from years ~10 to 18 of life and in mice from weeks ~3 to 8 of life, and is a period of intense somatic maturation, including sexual development (*Bell, 2018*), and that correlates with a decrease in impulsive behavior characteristic of the juvenile phase (*Rosati et al., 2023*; *Doremus-Fitzwater et al., 2012*; *Hammond et al., 2011*; *Konstantoudaki et al., 2018*).

The maturation of the mPFC includes structural and functional modifications during childhood and adolescence (*Chini and Hanganu-Opatz, 2021*; *Sakurai and Gamo, 2019*). Although it has been long hypothesized that the neural changes occurring in the mPFC during development are central to the emergence of behavioral control (e.g. *Durston and Casey, 2006*; *Sowell et al., 1999*) the specific plastic arrangements underlying behavioral control development remain poorly understood.

A number of studies have focused on the local changes of mPFC circuits, such as the changes in cortical thickness caused by cellular structural plasticity and synaptic pruning that are characteristic of early postnatal developmental phases in both humans and mice (*Nagy et al., 2004*; *Alexander-Bloch et al., 2013*; *Ueda et al., 2015*; *Kolk and Rakic, 2022*), as a putative locus underlying cognitive development. More recently, studies in rodents have shed light on the development of long-range top-down mPFC extracortical connections as the putative origin of certain aspects of cognitive development (*Klune et al., 2021*). In particular, the development of mPFC afferents to the amygdala may shape the response to threats across different stages of development (*Arruda-Carvalho et al., 2017*; *Dincheva et al., 2015*; *Gee et al., 2016*), and the development of mPFC input onto the dorsal raphe nucleus (DRN) shapes the response to stress (*Soiza-Reilly et al., 2019*).

A growing body of evidence supports that 5-hydroxytryptamine (5-HT) neuron activity in the DRN is related to increases in the ability to wait for rewards (*Winstanley et al., 2005*; *Fonseca et al., 2015*; *Lottem et al., 2018*; *Miyazaki et al., 2011*; *Miyazaki et al., 2018*; *Miyazaki et al., 2014*). This kind of behavior can be considered an instance of behavioral or cognitive control (*Cools et al., 2008*; *Dayan and Huys, 2009*), which can be manifested either as passive waiting (*Fonseca et al., 2015*; *Miyazaki et al., 2012*; *Miyazaki et al., 2018*; *Miyazaki et al., 2014*) or active persistence (*Lottem et al., 2018*). The raphe has likewise been implicated in the active overcoming of adverse situations (*Nishitani et al., 2019*; *Ohmura et al., 2020*; *Warden et al., 2012*).

The mPFC sends a dense glutamatergic projection to the DRN at the adult stage (*Pollak Dorocic et al., 2014*; *Weissbourd et al., 2014*; *Zhou et al., 2017*), which can bidirectionally modulate the activity of 5-HT DRN neurons through monosynaptic excitation or disynaptic feedforward inhibition through local interneurons (*Challis et al., 2014*; *Geddes et al., 2016*; *Maier, 2015*; *Warden et al., 2012*). Selective optogenetic activation of the mPFC inputs to the DRN elicits active behavioral responses in a challenging context (*Warden et al., 2012*), and perturbations in the development of this pathway lead to maladaptive anxiety levels (*Soiza-Reilly et al., 2019*). Conversely, optogenetic

activation of the DRN 5-HT input to the mPFC specifically promotes waiting for probabilistic rewards (*Miyazaki et al., 2020*).

Given the reciprocal connectivity between the mPFC and DRN (*Puig and Gulledge, 2011*) and that both areas causally modulate animals' ability to wait for delayed rewards (*Ciaramelli et al., 2021*; *Fonseca et al., 2015*; *Miyazaki et al., 2011*; *Miyazaki et al., 2018*; *Murakami et al., 2017*; *Schweig-hofer et al., 2008*), it seems plausible that the maturation of mPFC input to the DRN over develop-ment could underlie the emergence of behavioral persistence in mice.

Therefore, we sought to characterize the development of cortical innervation onto the DRN and its functional consequences in the context of behavioral persistence. In contrast to previous studies in which adult behavioral readouts were assessed after developmental perturbations (*Bitzenhofer et al., 2021*; *Soiza-Reilly et al., 2019*), we undertook a longitudinal study, characterizing behavior, synaptic physiology and anatomy, in parallel, from adolescence to adulthood. First, using a trans-genic line (Rbp-Cre) that targets the layer 5 neurons that provide the neocortical input to the DRN, we discovered that this input undergoes a dramatic increase in potency over the course of develop-ment from 3 to 4 weeks (juvenile) to 7 to 8 weeks (adult). Then, using a probabilistic foraging task, we found that mice's behavior persistence increased over the same period. Finally, using a genetic ablation technique that primarily affected the mPFC, we showed that ablation of neocortical input to the DRN in adult mice recapitulated the juvenile foraging behavior. Together, these results identify a descending neocortical pathway to the DRN that is critical to the maturation of behavioral control that characterizes adulthood.

## Results

### Cortical top–down input over the dorsal raphe matures in the transition between adolescence and adulthood in mice

First, to characterize the development of neocortical projections to the DRN, we focused on the affer-ents of layer 5 neurons, which are the primary origin of these projections (*Pollak Dorocic et al., 2014*). To do so, we used a mouse line expressing channelrhodopsin-2 (ChR2) under the *Rbp4* promoter (Rbp4-Cre/ChR2-loxP) which targets both intra- and extracortical projecting layer 5 neurons (*Leone et al., 2015*; *Gerfen et al., 2013*; *Tervo et al., 2016*) and that has been previously used to map the postnatal development of extracortical projections (*Peixoto et al., 2016*). Importantly, this approach represents key advantages over alternative viral based strategies as it is insensitive to injection size and location and avoids surgeries in pup mice, thus not introducing an unwanted source of early life stress (*Ririe et al., 2021*).

We performed ChR2-assisted circuit mapping (CRACM) (*Petreanu et al., 2009*) of cortical affer-ents in brain slices containing the DRN obtained from Rbp4/ChR2 mice between postnatal weeks 3 and 12 (*Figure 1A*). Taking advantage of the fact that ChR2-expressing axons are excitable even when excised from their parent somata, we evoked firing of presynaptic ChR2-expressing cortical axons innervating the DRN while recording the electrophysiological responses of postsynaptic DRN neurons. We assessed the fraction of recorded DRN neurons receiving cortical excitatory synaptic input (connection probability, Pcon) and the strength of this connection (amplitude of the evoked synaptic response) at different developmental time points.

We found a dramatic increase in the connection probability and amplitude of cortico-raphe input between weeks 3 and 8 (*Figure 1B, C*). At 3–4 weeks (juvenile mice), the probability of DRN neurons receiving cortical input was equal to 0.07. This probability increased significantly to 0.66 (Pcon 3–4 weeks vs. Pcon 5–6 weeks, Chi-square test $\chi^2$ (1, $N$ = 52 neurons) = 24.1, p = 0.00001) at weeks 5–6, reaching a peak connection probability of 0.82 at weeks 7–8 (*Figure 1C*). Between 5–6 and 7–8 weeks (i.e. late juvenile to adult mice), the amplitude of the optogenetically evoked currents increased from 27.3 ± 6.2 to 128 ± 15.7 pA (mean ± standard error of the mean [SEM], two-tailed $t$-test, $t$(55) = 4.03, p = 0.002). To test whether there is a further development of this pathway in the later stages of development, we recorded slices from 5 to 6 months old mice. We observed no further increase in either the connection probability (Pcon 7–8 weeks = 0.82 vs. Pcon 5–6 months = 0.80, Chi-square test $\chi^2$ (1, $N$ = 70 neurons) = 0.03, p = 0.84) or the input magnitude (7–8 weeks old = 126 ± 15 pA vs. 5–6 months old = 113 ± 14.1 pA, two-tailed $t$-test, $t$(55) = 1.27, p = 0.21, *Figure 2B, C*). Altogether, these results suggest that the cortico-raphe pathway gradually matures between weeks

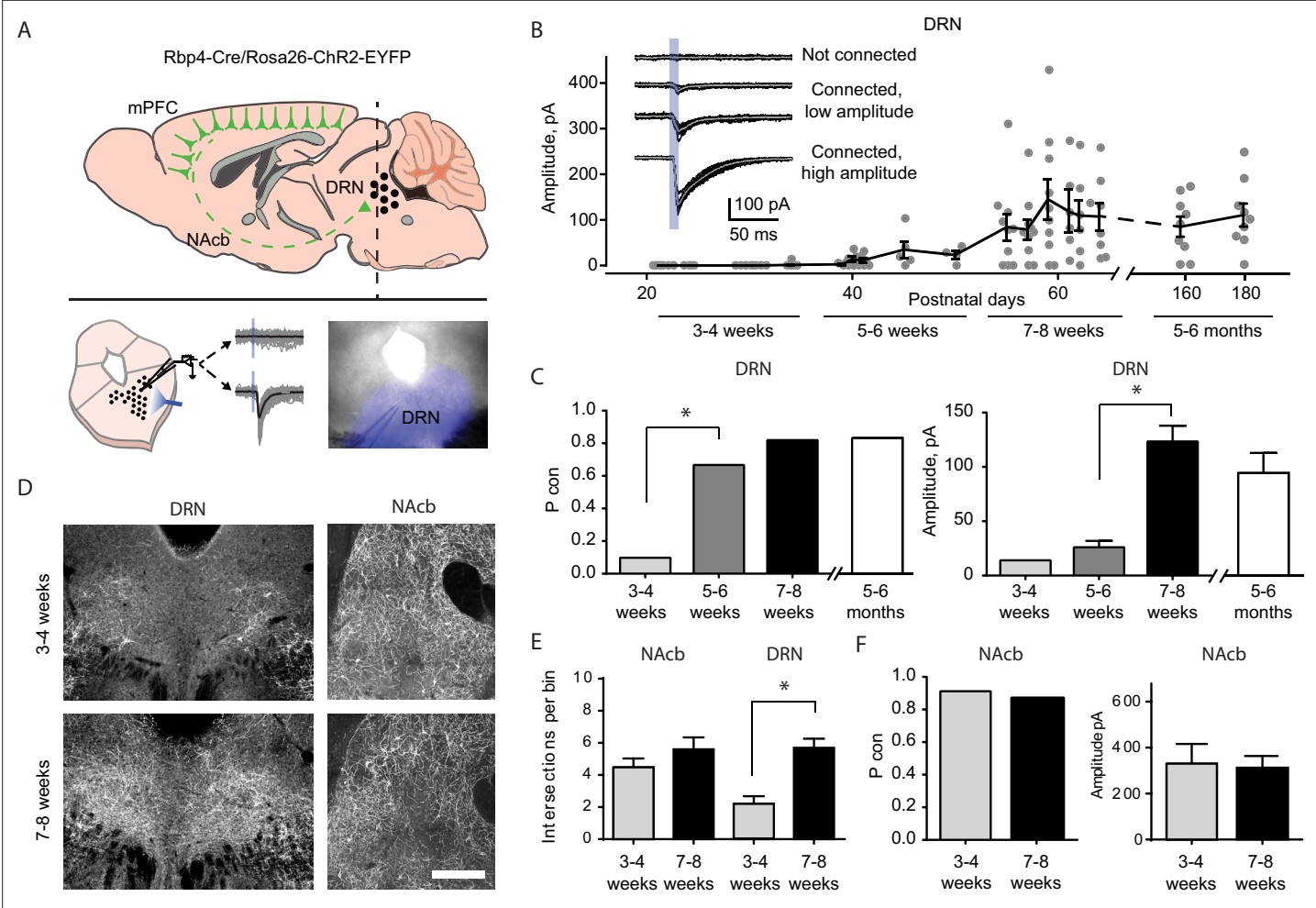

**Figure 1.** Top–down cortico-raphe connections develop over adolescence in mice. (**A**) Schematic representation of a sagittal view of an Rbp4-ChR2 mouse brain illustrating top–down cortico-raphe afferents. Coronal slices containing the dorsal raphe nucleus (DRN) were obtained ex vivo, and whole-cell recordings of DRN neurons were performed to assess cortical connectivity upon light stimulation. (**B**) Optogenetically-evoked excitatory postsynaptic currents (EPSCs) were recorded in DRN neurons contacted by ChR2-expressing cortical axons (122 neurons, 20 Rbp4-ChR2 mice). The current amplitude of cortico-raphe connections is plotted as a function of postnatal age in mice. (**C**) Pooled connection probability (connected cells/total cells) and averaged connection amplitude of cortico-DRN afferents at four different developmental points: early juvenile (3–4 weeks), late juvenile (5–6 weeks), early adult (7–8 weeks), and late adult (5–6 months). (**D**) Example images illustrate an increased cortico-DRN innervation in adult mice compared to juveniles, while the cortico-accumbens innervation remains constant over the same time period. Scale bar = 400 μm. (**E**) Number of axonal intersections quantified in the DRN and nucleus accumbens of juvenile and adult mice. (**F**) Pooled connection probability and averaged connection amplitude of cortico-accumbens afferents in early juvenile and early adult mice. *p < 0.05.

The online version of this article includes the following source data and figure supplement(s) for figure 1:

**Source data 1.** Electrophygiological recordings and axonal quantification.

**Figure supplement 1.** Changes in cortico-raphe connectivity over development are not explained by changes in the location of the recorded dorsal raphe nucleus (DRN) neurons.

**Figure supplement 1—source data 1.** Electrophysiological properties of DRN neurons and optogenetic controls.

**Figure supplement 2.** Changes in cortico-raphe connectivity over development are not explained by changes in membrane properties of dorsal raphe nucleus (DRN) neurons or by differential ChR2 expression of ChR2 under the *Rbp4* promoter over time.

3 and 8 and then plateaus. Importantly, the location of the recorded DRN neurons was comparable between juvenile and adult mice (*Figure 1—figure supplement 1*) and thus, the connectivity changes observed across development do not reflect a biased sampling of differentially innervated subregions of the DRN. Furthermore, the passive electrical properties did not change over development, as measured by input resistance (3–4 weeks: median = 444 MΩ, 95% confidence interval [CI] = [370, 676],

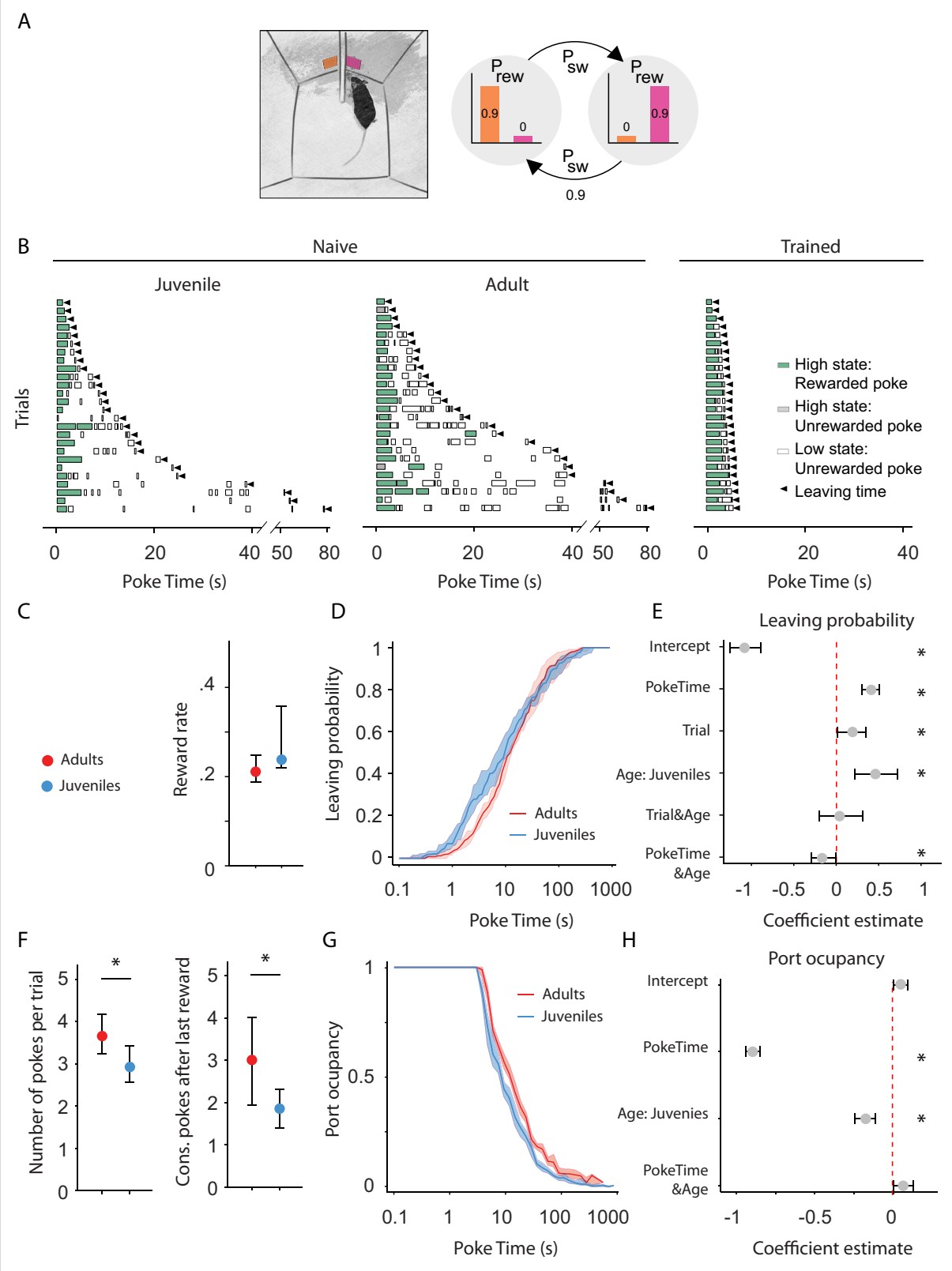

**Figure 2.** Adult mice persist longer than juveniles in exploiting a foraging patch. (**A**) Illustration of the rodent foraging task. Water-deprived mice seek rewards by probing two nose-ports. (**B**) Randomly selected examples of poking behavior throughout a naive juvenile, naive adult, and trained adult behavioral session sorted by trial length. Pokes in the active state can be rewarded (in green) or not (in gray). Pokes in the inactive state are never rewarded (in white). After the state switches, the mice have to travel to the other side (left or right port, L annd R) to obtain more water. Leaving time

*Figure 2 continued on next page*

*Figure 2 continued*

is illustrated with black triangles. (**C**) Median ± 95% confidence interval (CI) of the reward rate per second for juvenile and adult mice. (**D**) Cumulative distribution of the probability of leaving (median ± 95% CI across mice) as time elapses from the first poke in a trial for adults and juvenile animals. (**E**) Regression coefficients ± 95% CI resulting from a parametric bootstrap (*n* = 1000) of a mixed models logistic regression to explain the probability of leaving. * indicates predictors with a significant impact on the probability of leaving. (**F**) Median ± 95% CI of the number of pokes per trial (left) and the consecutive pokes after the last reward (right). Juvenile mice do a significantly lower amount of pokes per trial and pokes after the last reward compared to adult mice. (**G**) Port occupancy as a function of trial time elapsed for juveniles and adults. (**H**) Regression coefficients ± 95% CI resulting from a parametric bootstrap (*n* = 1000) of a mixed models logistic regression to explain the port occupancy, as in E. All analyses in C–H computed by pooling the data from all sessions of juvenile (*N* = 21) or adult (*N* = 23) mice, yielding a total of 2875 trials (juveniles = 1347, adults = 1528) and 9596 pokes (juveniles = 3908, adults = 5688).

The online version of this article includes the following source data and figure supplement(s) for figure 2:

**Source data 1.** Foraging task behavior over development.

**Figure supplement 1.** Description of mouse nose foraging behavior over the session progression and according to sex.

**Figure supplement 1—source data 1.** Behavioral controls across experiments.

5–6 weeks: median = 612 MΩ, 95% CI = [402, 925], 7–8 weeks: median = 731 MΩ, 95% CI = [519, 943], 5–6 months: median = 532 MΩ, 95% CI = [385, 664], Kruskal–Wallis $H(3)$ = 6.06, p = 0.11) and input capacitance (3–4 weeks: median = 20.7 pF, 95% CI = [17.4, 25.9], 5–6 weeks: median = 22.8 pF, 95% CI = [15.9, 24.8], 7–8 weeks: median = 20.8 pF, 95% CI = [18.3, 28.5], 5–6 months: median = 23.5 pF, 95% CI = [14.1, 44.7], Kruskal–Wallis $H(3)$ = 0.81, p = 0.84) (***Figure 1—figure supplement 2A, B***), suggesting that changes in the passive propagation of current through DRN neurons is not the underlying cause of the apparent increase in connection probability and input magnitude observed over time.

In these experiments, the onset of ChR2 expression is dictated by the Cre recombinase expression under the control of the native *Rbp4* promoter over development. Therefore, if in the juvenile cortex there were fewer neurons expressing *Rbp4* or the onset of expression was near our recording time point, this could affect the net amount of ChR2-expressing top–down cortical axons and/or their net excitability. To control that our findings reflect a development process and not a genetic artifact caused by the temporal dynamics of *Rbp4* expression, we performed two additional control experiments in one of the main cortical origins of afferents onto the DRN, the mPFC (***Weissbourd et al., 2014***; ***Zhou et al., 2017***).

First, we compared the density of neurons in the mPFC expressing a fluorescent reporter (tdTomato) under the control of the *Rbp4* promoter in juvenile and adult mice. The same density of tdTomato expressing somas was detected in the mPFC of juvenile and adult Rbp4-Cre/tdTomato-loxP mice (Juveniles: median = 4.59 somas per 0.01 mm$^2$, 95% CI = [4.59, 6.18], vs. Adults: median = 5.22 somas per 0.01 mm$^2$, 95% CI = [4.35, 6.06], Mann–Whitney *U* test (*N* Juveniles = 3, *N* Adults = 4) = 6, p = 0.99, ***Figure 1—figure supplement 2C***), indicating that a comparable number of neurons underwent a Cre-dependent recombination of the tdTomato fluorescent reporter under the control of the *Rbp4* promoter at both developmental time points. Second, we compared the light-evoked somatic current produced in layer 5 neurons expressing ChR2 under the *Rbp4* promoter in juvenile and adult mice. In agreement with the previous control, layer 5 neurons in the mPFC expressing ChR2 under the *Rbp4* promoter produced the same amount of photocurrent upon light stimulation in juvenile and adult mice (repeated measures analysis of variance [ANOVA], $F(1, 11)$ = 0. 138, p = 0.71 for age factor, ***Figure 1—figure supplement 2D***). These experiments show that juvenile and adult mice have similar densities of cortical layer 5 projection neurons that could give rise to DRN afferents and that these neurons express similar amounts of ChR2 and thus, if present, projections should be equally detectable by optogenetic circuit mapping across ages.

To further understand the mechanisms underlying the cortico-raphe input strengthening observed over development, we investigated whether the changes of connection probability and input amplitude we observed were accompanied by differences in the density of cortical axonal innervation over the DRN. Indeed, we observed a significantly higher density of *Rbp4* positive axons around the DRN in adults compared to juveniles (2.2 ± 0.47 vs. 5.7 ± 0.56 axons per bin in juvenile vs. adult mice, two-tailed *t*-test (*N* Juveniles = 4, *N* Adults = 7), $t(9)$ = 4.15, p = 0.002, ***Figure 1D, E***). This observation supports the idea that the increase in physiological strength we observed reflects in part the growth of new connections between the neocortex and DRN.

To assess whether the development of cortico-raphe projections is specific to raphe projecting cortical afferents or it reflects a more general maturation of corticofugal projections over adolescence in mice, we mapped the anatomical and synaptic development of cortico-accumbens projections, which mainly originate in the mPFC (*Phillipson and Griffiths, 1985*; *Li et al., 2018*) and whose functional connectivity has been previously assessed in juvenile rodents (*Gorelova and Yang, 1996*). In contrast to cortico-raphe afferents, cortico-accumbens projections did not undergo any significant structural change over the same developmental period (4.5 ± 0.54 vs. 5.8 ± 0.74 axons per bin in juvenile vs. adult mice, two-tailed *t*-test (*N* Juveniles = 3, *N* Adults = 7), *t*(8) = 1.09, p = 0.40, *Figure 1D, E*). Consistent with the anatomy, the ChR2-assisted mapping of cortico-accumbens connections in juvenile and adult Rbp4-ChR2 mice revealed no change in either the connection probability (Pcon 3–4 weeks = 0.90 vs. Pcon 7–8 weeks = 0 .87, Chi-square test $\chi^2$ (1, *N* = 19 neurons) = 0.03, p = 0.81) or the input amplitude in the transition from juveniles to adults (two-tailed *t*-test, *t*(15) = 0.15, p = 0.88, *Figure 1F*). Altogether, these observations reveal the structural and synaptic development of a subpopulation of cortical afferents targeting the DRN during the transition to adulthood in mice that does not reflect a generalized development of corticofugal projections.

## Baseline persistence correlates with the maturation of cortico-raphe input in the transition between adolescence and adulthood in mice

To investigate the development of behavioral persistence in mice, we employed a self-paced probabilistic foraging task (*Vertechi et al., 2020*). The setup consists of a box with two nose-ports separated by a barrier (*Figure 2A*). Each nose-port constitutes a foraging site that water-deprived mice can actively probe in order to receive water rewards. Only one foraging site is active at a time, delivering reward with a fixed probability. Each try in the active site can also cause a switch of the active site's location with a fixed probability (*Figure 2B*). After a state switch, mice have to travel to the other port to obtain more reward, bearing a time cost to travel. In this task, a trial is defined as a bout of consecutive attempts on the same port, before leaving, and the amount of time spent attempting to obtain reward in one port before switching is the primary measure of persistence, independent of the specific strategy used by the mice (see Discussion).

We compared the behavior of juvenile (weeks 3–4) and adult mice (weeks 7–8) on their first exposure to the apparatus and task, ensuring that differences in persistence do not arise from differences in learning about the task. We used an environment characterized by high reward probability ($p_{rwd}$ = 90%) and high site-switching probability ($p_{sw}$ = 90%) (*Figure 2A*). These statistics produce a small number of rewards per trial (Rewards per trial: minimum = 0, maximum = 3) (*Figure 2B*), and maximize the number of trials performed in one session. Both groups obtained a comparable reward rate (*Figure 2C*; Adults: median = 0.02 rewards per second, 95% CI = [0.0020, 0.003], Juveniles: median = 0.023 rewards per second, 95% CI = [0.001, 0.012]; Mann–Whitney *U* test (*N* Adults = 21, *N* Juveniles = 23) = 82.0, p = 0.16), indicating that juveniles and adults do not differ in terms of overall competence in performing the task.

Compared to well-trained animals, naive mice tended to interleave poking in the port with investigating the apparatus. Presumably due to the novelty of the environment, they often took long pauses in between pokes at the same port. This resulted in a less regular poking structure than experienced mice (*Figure 2B*). In light of this observation, it is unlikely that animals were tracking the number of foraging attempts executed on the same site. Therefore, we tested whether leaving choices were better explained as a function of elapsed time. Indeed, we found that contrary to trained mice (*Vertechi et al., 2020*), for naive mice, elapsed time has more explanatory power on leaving decisions (Akaike information criterion test Leave~1 + PokeTime + (1 + PokeTime|MouseID) vs. Leave~1 + PokeNumber + (1 + PokeNumber|MouseID): $AIC_{time}$ = 1.13e$^4$, $AIC_{number}$ = 1.14e$^4$ p: 1.94e$^{-23}$).

To test how postnatal development affects innate persistence, we performed a logistic regression for probability of leaving the patch as a function of the time elapsed within the trial (*Figure 2C*), trials completed within the session, and the age of the animal (juvenile vs. adult):

Leave~1 + PokeTime + Trial + Age + PokeTime&Age + Trial&Age + ( 1 + PokeTime + Trial|Mouse).

We accounted for the individual variability through generalized linear mixed models with random intercept and slope for each mouse (see Methods for the implementation). A factor was considered to significantly affect the decision to leave if the value of its estimated coefficient plus 95% CI (1000 parametric bootstrap analysis, see Methods) did not cross 0.

This analysis showed that juveniles are significantly more likely to leave earlier than adults and thus, less persistent (*Figure 2D, E* and *Figure 2—figure supplement 1A*). Consistently, including the Age group in the model significantly contributes to the ability to explain leaving decisions (likelihood ratio test on Leave~1 + PokeTime + Trial + Age + PokeTime&Age + Trial&Age + (1 + PokeTime + Trial|MouseID) vs. Leave~1 + PokeTime + Trial + (1 + PokeTime + Trial|MouseID): $X^2_{(3)}$ = 15.65, p = 0.0013).

Another factor significantly explaining foraging was the amount of trials performed. We found that throughout the session animals were progressively leaving earlier — the probability of leaving increased as a function of Trial, indicating that animals become less persistent over the course of the session (*Figure 2—figure supplement 1B*, likelihood ratio test on Leave~1 + PokeTime + Trial + Age + PokeTime&Age + Trial&Age +(1 + PokeTime + Trial|MouseID) vs. Leave~1 + PokeTime + Age + PokeTime&Age + (1 + PokeTime|MouseID): $X^2_{(-5)}$ = 26.88, p < 1e$^{-5}$). This is likely reflecting a drop in motivation due to the water drunk during the session or accumulated tiredness. Differences in any of these aspects could explain the change in persistence, for instance juveniles might get satiated or tired faster than adults. To test this possibility we analyzed whether the behavioral change over trials was different between the two groups. Crucially, there was no significant interaction between Trial and Age factors (*Figure 2E*), indicating that differences in satiety or fatigue accumulated throughout the session do not underlie the change in persistence between juveniles and adults.

Although the time elapsed from the beginning of a trial is a better metric to explain leaving decisions, it does not distinguish between active persistence and spurious pauses in poking. Changes in leaving time could be caused either by an increase in the number of attempts performed or by an increase in the time between attempts. We therefore compared the number of attempts per trial in adults and juveniles, both as overall number of pokes and as consecutive unrewarded pokes performed after the last reward (*Vertechi et al., 2020*; *Figure 2F*). Adults made more pokes per trial than juveniles (*Figure 2F*; Adults: median = 3.65 pokes per trial, 95% CI = [0.42, 0.51], Juveniles: median = 2.92 pokes per trial, 95% CI = [0.35, 0.50]; Mann–Whitney *U* test (*N* Adults = 21, *N* Juveniles = 23) = 354.5, p = 0.008, effect size = 0.73), and more consecutive failures after the last reward *Figure 2F*; Adults: median = 2.69 pokes after last reward, 95% CI = [0.39, 0.53], Juveniles: median = 1.96 pokes after last reward, 95% CI = [0.41, 0.52]; Mann–Whitney *U* test (*N* Adults = 21, *N* Juveniles = 23) = 351.0, p = 0.009, effect size = 0.73. Next, we assessed the temporal profile of poking behavior in terms of port occupancy—quantified as the cumulative time spent with the snout in the port divided by the overall time elapsed from the trial beginning (*Figure 2G*). We found that adult animals were consistently poking for longer than juveniles, as indicated by both a significant effect of the Age variable (*Figure 2H*) and significant improvement in explanatory power with the Age factor: likelihood ratio test on Occupancy~1 + PokeTime + Age + Age&PokeTime + (1 + PokeTime|MouseID) vs. Occupancy~1 + PokeTime +(1 + PokeTime|MouseID), $X^2_{(2)}$ = 20.10, p < 1e$^{-4}$. Furthermore, the difference between the two groups was constant throughout the trial, as shown by the lack of a significant interaction between poke time and age factors (*Figure 2H*). Altogether these results indicate that adult animals are actively more persistent than juveniles and this is not explained by qualitative differences in their poking behavior.

Finally, to assess whether the observed change in persistence is consistent in male and female mice we tested, in the same mice cohort, whether the variable sex played a role in leaving decisions. We found no significant effect of Sex on the probability of leaving either alone or in interaction with animals' age (*Figure 2—figure supplement 1C*), and including Sex as a predictor had no significant improvement in the model's ability to explain the decision to leave (*Figure 2—figure supplement 1C*, likelihood ratio test on Leave~1 + PokeTime + Trial + Age + Sex + Age&Sex + (1 + PokeTime + Trial|MouseID) vs. Leave~1 + PokeTime + Trial + Age + (1 + PokeTime + Trial|MouseID): $X^2_{(2)}$ = 3.46, p = 0.18). These results indicate that the maturation of persistence occurs at a similar rate in male and female mice.

## Cortico-DRN pathway ablation in adult mice recapitulates juvenile behavioral features

The above results establish a correlation between the development of the descending cortical input to the DRN and the emergence of behavioral persistence. To test the causal link between the development of cortico-raphe afferents and the observed increase in persistence during foraging, we next ablated the cortico-raphe pathway in adult mice and assessed the impact on behavioral persistence.

To ablate cortico-raphe afferents, we used an engineered version of Caspase3 (taCasp3-TEVp) that is able to trigger apoptosis bypassing cellular regulation upon activation by the TEV protease, which is coexpressed in the same construct (*Yang et al., 2013*). We packaged a Cre-dependent taCasp3-TEVp construct (or the reporter tdTomato as a control) in a retrogradely traveling AAV vector (rAAV), that we locally delivered in the DRN of Rbp4-Cre mice (*Figure 3—figure supplement 1*). This approach resulted in the fluorescent tagging of cortico-raphe layer 5 projecting neurons in control mice (tdTomato mice) and in the ablation of the same corticofugal pathway in taCasp3-TEVp-injected mice (Caspase mice) (*Figure 3A* and *Figure 3—figure supplement 1*).

We found tdTomato+ expressing neurons in five cortical areas (*Figure 3—figure supplement 2A–G*), which is largely consistent with previous reports (*Xu et al., 2021*). Furthermore, these neurons mainly originated in the prefrontal cortex, being 13 times more abundant than those from other cortical areas outside the prefrontal cortex. The prelimbic/infralimbic (PL/IL) and anterior cingulate (AC) cortices, which constitute the mPFC, were the areas with the highest density of DRN-projecting tdTomato+ somas in control animals (*Figure 3—figure supplement 2B–E*, median = 3.41 neurons per layer 5 bin, 95% CI = [1.45, 3.81] for PL/IL and median = 1.54 neurons per layer 5 bin, 95% CI = [1.18, 3.64] for AC) and consistently more extensive neuron density loss in caspase-injected mice, quantified using the pan-neuronal marker NeuN (*Figure 3A–C*, control $n$ = 8 mice vs. caspase $n$ = 7 mice, two-sample Kolmogorov–Smirnoff test = 0.028, p = 0.002 for PL/IL and $D$ = 0.024, p = 0.01 for AC). We also found tdTomato+ somata in the medial orbitofrontal cortex (MO) of the control group; however, this projection was less robust in terms of tdTomato+ labeled neurons across animals (*Figure 3—figure supplement 2B, C*, median = 1.34 neurons per layer 5 bin, 95% CI = [0.37, 4.81]) and, consistently, the difference in layer 5 NeuN densities between control and caspase mice was not significant (*Figures 3C and 5*, $D$ = 0.017, p = 0.08).

Apart from the mPFC, sparse labeling of tdTom+ neurons was found in more posterior levels of the neocortex, namely in the retrosplenial cortex (RS) and in the temporal association cortex (TeA) (*Figure 3—figure supplement 2B, F, G*; median = 0.11 neurons per layer 5 bin, 95% CI = [0.0, 0.56] for Rs and median = 0.0 neurons per layer 5 bin, 95% CI = [0.0, 0.52] for TeA). However, it is worth noting that tdTom+ neurons were only found in the RS of 5 out of 8 control animals, and in the TeA of 3 out of 8 control animals. Consistently, the reduction in NeuN layer 5 neuronal density in these two areas was minimal and non-significant compared to controls (*Figure 3C*, $D$ = 0.034, p = 0.12 for RS and $D$ = 0 .025, p = 0.19 for TeA). In addition, no differences in NeuN density were observed between caspase- and tdTomato-injected animals in an area that does not contain tdTomato expressing somas and therefore not projecting to the DRN which serves as a negative control to rule out unspecific biases in our quantification method (M1, *Figure 3C*, $D$ = 0.019, p = 0.15). These observations suggest that our ablation approach primarily affected mPFC–DRN-projecting neurons, particularly from the PL/IL and AC cortices.

When investigating the distribution of tdTomato expressing somas, we observed weak collateral projections of the cortical subpopulation projecting to the DRN in the lateral septum, lateral hypothalamic nucleus, the ventral tegmental area and the anterior periaqueductal gray; medium collateral axonal density in anterior subcortical olfactory nuclei (anterior dorsal endopirifrm, anterior olfactory nucleus, dorsal taenia tecta, and islands of Calleja) and the substantia nigra; and heavy collateralization in the dorsomedial striatum (*Figure 3—figure supplement 3*).

We then assessed the impact of ablation of cortical input to the DRN on behavioral persistence using the same foraging paradigm and analysis we adopted to assess persistence in adults and juveniles (*Figure 3D*). First, we confirmed that both groups could perform the task comparatively well, obtaining a similar reward rate (*Figure 3E*; tdTomato: median = 0.033 rewards per second, 95% CI = [0.019, 0.003], Caspase: median = 0.034 rewards per second, 95% CI = [0.010, 0.008]; Mann–Whitney $U$ test ($N$ tdTomato = 8, $N$ Caspase = 7) = 21.0, p = 0.46). However, in line with our observations in juvenile mice, we observed that Caspase mice made significantly fewer pokes per trial (*Figure 3E*; tdTomato: median = 3.99 pokes per trial, 95% CI = [1.15, 0.98], Caspase: median = 2.80 pokes per trial, 95% CI = [0.47, 0.54]; Mann–Whitney $U$ test ($N$ Caspase = 7, $N$ tdTomato = 8) = 49, p = 0.014, effect size = 0.88) and significantly lower number of pokes after last reward (*Figure 3E*; tdTomato: median = 3.01 pokes after last reward, 95% CI = [1.16, 0.98], Caspase: median = 1.87 pokes after last reward, 95% CI = [0.49, 0.44]; Mann–Whitney $U$ test ($N$ Caspase = 7, $N$ tdTomato = 8) = 48, p = 0.02, effect size = 0.86) when compared to tdTomato controls. Furthermore, and similar to juvenile

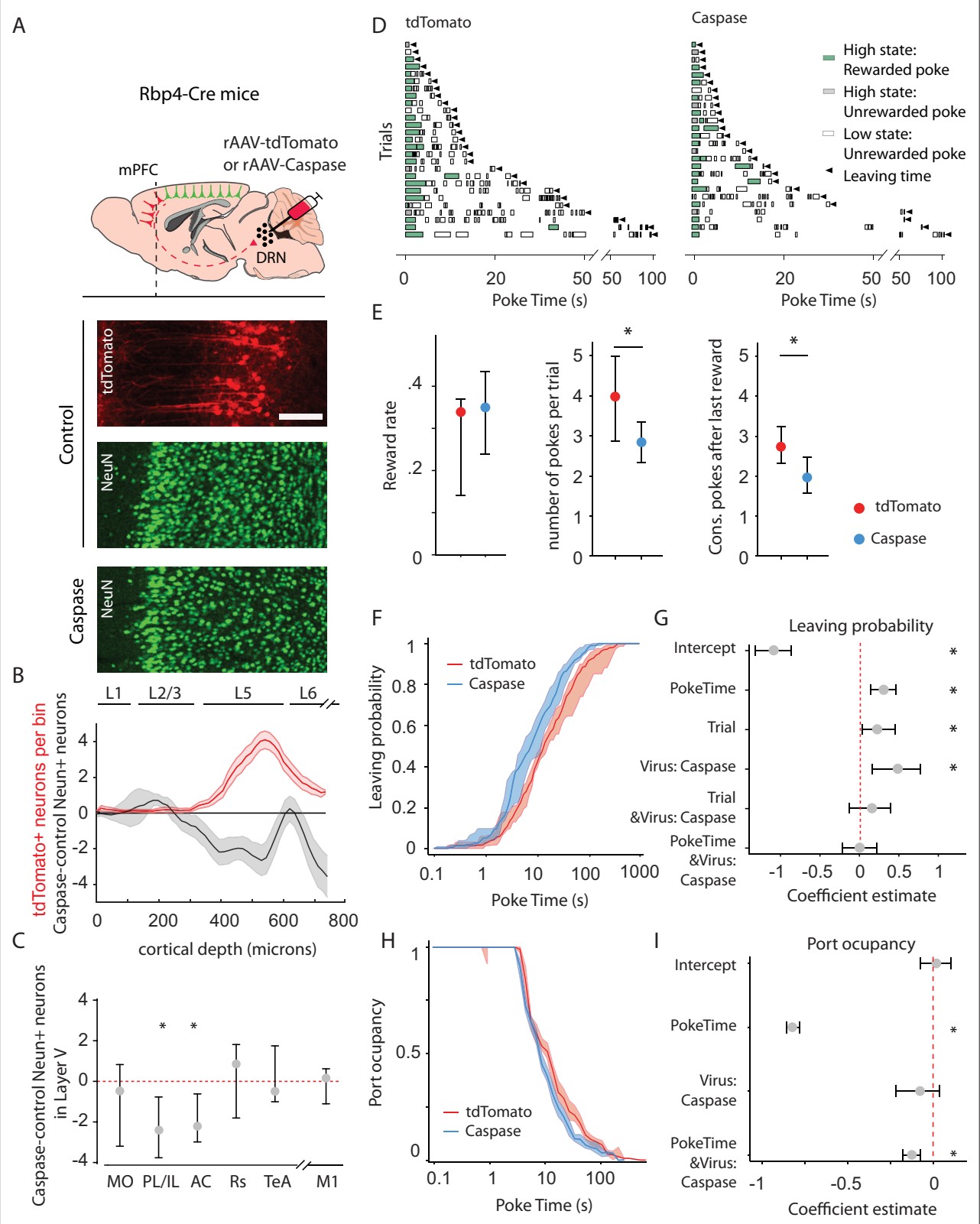

**Figure 3.** Animals lacking cortico-raphe projections show less behavioral persistence in exploiting a foraging patch. (**A**) Schematic representation of the ablation strategy used for behavioral assessment. Retrogradely transporting AAV vectors expressing either the fluorescent reporter tdTomato (rAAV-tdTomato) or the intrinsically active apoptosis triggering Caspase3 (rAAV-Caspase, Caspase group were locally delivered in the dorsal raphe nucleus (DRN) of Rbp4-Cre mice). In the cortical areas containing tdTomato expressing neurons in control animals (in example picture, PL/IL cortex) the density

*Figure 3 continued on next page*

*Figure 3 continued*

of neurons was quantified with an immunohistochemistry protocol against the pan-neuronal marker NeuN and compared to the neuronal densities obtained in the same cortical areas of ablated mice (scale bar = 200 microns). (**B**) Distribution of the neuronal density difference between ablated mice and the mean density of control mice per cortical depth bin in the PL/IL cortex (black shaded error plot). The neuronal density loss observed in ablated mice when compared to control NeuN densities matches the cortical depth in which tdTomato neurons are located (red shaded area). Shaded error plots represent mean ± standard error of the mean (SEM). (**C**) Summary of caspase-control NeuN density per brain area (MO: median = −0.46, 95% confidence interval [CI] = [−2.81, 1.06], PL/IL: median = −2.3, 95% CI = [−4.01, −1.44], AC: median = −2.26, 95% CI = [−3.59, −0.17], Rs: median = 0.75, 95% CI = [−1.80, 1.81], TeA: median = −0.63, 95% CI = [−1.01, 1.76], and M1: median = 0.14, 95% CI = [−1.81, 0.73]). * indicates p<0.05. (**D**) Randomly selected examples of poking behavior for a tdTomato and caspase behavioral session sorted by trial length. Pokes in the active state can be rewarded (in green) or not (in gray). Pokes in the inactive state are never rewarded (in white). Leaving time is illustrated with black triangles. (**E**) Median ± 95% CI fraction of the reward rate per second (left), pokes per trial (middle) and the consecutive pokes after the last reward (right). Caspase mice obtain a comparable number of rewards per trial but do a significantly lower amount of pokes per trial and pokes after the last reward compared to control tdTomato mice. * indicates p<0.05. (**F**) Cumulative distribution of the probability of leaving as a function of trial time elapsed (median ± 95% CI across mice) for tdTomato and Caspase animals. (**G**) Regression coefficients ± 95% CI resulting from a parametric bootstrap (*n* = 1000) of a mixed models logistic regression to explain the probability of leaving. * indicates predictors with a significant impact on the probability of leaving. (**H**) Port occupancy as a function of trial time elapsed for tdTomato and Caspase. (**I**) Regression coefficients ± 95% CI resulting from a parametric bootstrap (*n* = 1000) of a mixed models logistic regression to explain the port occupancy. All analyses in B–G computed by pooling the data from the histology and the first session of Caspase (*N* = 7) or tdTomato (*N* = 8) mice, yielding a total of 1464 trials (Caspase = 939, tdTomato = 525) and 4742 pokes (Caspase = 2555, tdTomato = 2187).

The online version of this article includes the following source data and figure supplement(s) for figure 3:

**Source data 1.** Cell loss and behavioral quantification in caspase and control mice.

**Figure supplement 1.** Viral injections in the dorsal raphe nucleus (DRN) provide localized cortico-DRN axonal feedback and do not affect neuronal density.

**Figure supplement 1—source data 1.** NeuN and tdTomato density in the DRN.

**Figure supplement 2.** Labeling of Rbp4-expressing dorsal raphe nucleus (DRN)-projecting neurons with rAAV-tdTomato is consistent with cell density loss in mice injected with rAAV-Caspase3.

**Figure supplement 2—source data 1.** NeuN and tdTomato densities across cortical areas.

**Figure supplement 3.** Dorsal raphe projecting cortical neurons have dense collateral projections to the striatum.

animals (*Figure 2—figure supplement 1A*), Caspase animals had a higher chance of leaving the port, as indicated by a leftward shift in the cumulative distribution of leaving times (*Figure 3F*). We therefore applied logistic regression analysis to test this difference and assess how this effect relates to the factors previously identified in the age comparison. Animals lacking cortico-raphe projections were significantly more likely to leave the patch earlier than control animals (*Figure 3F, G*). Congruently, including the Virus factor significantly improved the model performance (likelihood ratio test on Leave~1 + PokeTime + Trial + Virus + PokeTime&Virus + Trial&Virus +( 1 +P okeTime + Trial|MouseID) vs. Leave~1 + PokeTime + Trial + (1 +P okeTime + Trial|MouseID): $X^2_{(3)}$ = 10.83, p = 0.012).

In line with the previous results, the probability of leaving increased with the number of trials performed. As previously outlined, this factor captures the effect of long-running changes, such as satiety and tiredness. Crucially, these effects did not differ between caspase and tdTomato infected groups, as shown by the lack of interaction between the trial and group factors (*Figure 3G*). Interestingly, the reduced persistence of Caspase animals did not scale with the elapsed time as for the juveniles (lack of interaction effect PokeTime&Virus, *Figure 3G*).

As with the case of juvenile versus adult mice, Caspase mice showed a shift toward shorter port occupancy (*Figure 3H*). A regression analysis showed that the viral intervention impacted significantly port occupancy (likelihood ratio test on Occupancy~1 + PokeTime + Virus + Virus&PokeTime +( 1 + PokeTime|MouseID) vs. Occupancy~1 + PokeTime + (1 + PokeTime|MouseID), $X^2_{(2)}$ = 12 .69, p = 0.018), and the regression analysis showed that this was mainly due to a progressive reduction during the trial rather than a subtractive effect (significant PokeTime & Virus: Caspase, not Virus: Caspase alone, *Figure 3I*).

Together, these results show that turning off the cortical input to the DRN, which mostly originates in the mPFC, makes adult mice behave more like juvenile mice when they are performing the same foraging task. This suggests that mature cortico-DRN innervation may be necessary for adult mice to be persistent in their behavior, and this pathway may help mice learn to be persistent in their behavior.

# Discussion

In the present study, we described how the postnatal maturation of the cortical innervation over the DRN during adolescence is linked to the performance of a probabilistic foraging task. Over the same period of development, the cortico-raphe projections underwent a dramatic increase in potency and mice developed an increase in persistence in foraging behavior. Ablation of this pathway in adult mice recapitulated the features observed in the behavior of juvenile mice, supporting a causal relationship between the cortico-raphe input and behavioral persistence.

In a wide variety of species, including mice, adolescence corresponds to the emancipation from the parents (**Spear, 2000**), a period in which individuals need to develop or refine skills to become independent. This ethological scenario may explain the evolutionary selection of juvenile behavioral traits (**Sercombe, 2014**; **Spear, 2000**), such as increased impulsivity or high risk taking behavior (**Laviola et al., 2003**; **Sercombe, 2014**). However, the abnormal development of behavioral control over the intrinsic behavioral tendencies of juveniles may underlie aspects of the etiopathology of impulsive and addictive disorders in adult humans (**Reiter et al., 2016**, **Wong et al., 2006**). In line with pre-adolescent humans' lack of delay gratification ability (**Mischel et al., 1989**), and with studies assessing impulsive behavior in mice over development (**Sasamori et al., 2018**), we found that mice of 3–4 weeks of age tend to be less persistent than 7–8 weeks old mice in a probabilistic foraging task. In a previous study, we showed that adult mice are capable of performing the task by adopting an effective inference-based strategy (**Vertechi et al., 2020**) which involves tolerating a fixed number of consecutive failures after the last received reward, independent of the total number of rewards obtained in that trial. This strategy is optimal because the state switch probability is independent of the reward probability. However, before learning this strategy, mice use a simpler 'stimulus-bound' strategy in which the number of rewards received tends to increase persistence during a trial (**Vertechi et al., 2020**). Altogether, our observations suggest that naive juvenile and adult mice forage in a similar manner but utilize different values for outcomes, which may reflect their niche specializations.

From a neural perspective, maturational changes of cortical areas, including the mPFC, have been previously linked to the emergence of cognitive skills during development in primates and humans (**Luna et al., 2015**; **Nagy et al., 2004**; **Velanova et al., 2008**). Such changes result in an increased top–down behavioral control and increased functional connectivity with cortical and subcortical targets (**Hwang et al., 2010**) in the transition between childhood to adulthood. However, the specific contribution of long-range top–down cortical circuits and the cellular mechanisms underlying its development had not been previously investigated.

Here, using optogenetic-assisted circuit mapping we characterized the structural and functional development of cortico-raphe projections that take place over adolescence in mice. A recent report showed that a subpopulation of DRN-projecting mPFC neurons increases their axonal contacts over the DRN in an earlier phase of postnatal development (weeks 1–2) (**Soiza-Reilly et al., 2019**). We found a cortico-DRN connection strength at 5–6 weeks postnatal similar to that reported by Soiza-Reilly and colleagues at a similar developmental time point (4–5 weeks of age). In addition, we found a connection strength at 7–8 weeks of age similar to those reported in adult rodents elsewhere (**Zhou et al., 2017**; **Geddes et al., 2016**). Thus, our findings are consistent with previous observations in the literature and suggest that the maturation of cortico-DRN afferents starts early in postnatal development and undergoes an extended development period, plateauing only after reaching 7–8 weeks of age. Among the previous studies investigating the postnatal development of top-down afferents from the neocortex in rodents (**Klune et al., 2021**; **Peixoto et al., 2016**; **Ferguson and Gao, 2014**), the latest afferent maturational process reported is the mPFC innervation over the basolateral amygdala, which occurs up to week 4 (**Arruda-Carvalho et al., 2017**). Thus, to our knowledge, the cortical innervation of the DRN represents the latest top–down pathway to develop.

Importantly, we found that the structural development of cortico-DRN projections is causally linked to the maturation of behavioral persistence in adult mice. Using a genetically-driven ablation approach (**Yang et al., 2013**), we selectively eliminated layer 5 cortical neurons projecting to the DRN in adult mice. The procedure resulted in a behavioral phenotype that recapitulated key features of the juvenile foraging behavior. We observed a reduction in behavioral persistence. This difference in behavioral persistence was small but reliable, and of similar magnitude (but, as expected, in the opposite direction) to the difference observed when optogenetically stimulating 5-HT DRN neurons in mice performing a probabilistic foraging task (**Lottem et al., 2018**). Furthermore, we localized the

origin of these projections and quantified the local neuronal loss. The PL, IL, and AC cortices, areas that comprise the so-called mPFC (*Klune et al., 2021*), suffered a significant loss with the procedure. Although we can not rule out the contribution of the other affected areas (Rs and TeA) in our caspase manipulation experiment, it is very likely that the areas with higher neuronal loss (mPFC) made a pivotal contribution to the behavioral changes we observed. It should be noted that the extent of layer 5 neurons affected by the caspase ablation in these cortical areas will be defined by the total percentage of layer 5 neurons expressing *Rbp4*. A previous study has shown that a Cre-dependent fluorescent reporter expressing retroAAV injected in the basal pontine nuclei of Rbp4-Cre mice produces a comparable density of labeled layer 5 cortical neurons as obtained with a standard retrograde tracer such as fluorogold (*Tervo et al., 2016*). This suggests that, at least for the case of cortico-pontine projection neurons, the *Rbp4* promoter grants genetic access to virtually all layer 5 projecting neurons. However, we cannot conclude that this holds true for the case of cortico-raphe projections and therefore future work will have to assess whether additional non-*Rbp4* populations of projecting neurons in these, or other cortical areas, contribute as well to the development of behavioral persistence.

Previous reports have shown that the pharmacological inactivation of the IL cortex reduces response persistence in a foraging task (*Verharen et al., 2020*). Moreover, lesions of the IL cortex improves performance on a reversal learning task (*Ashwell and Ito, 2014*), which resembles the increased behavioral flexibility observed in juvenile mice (*Johnson and Wilbrecht, 2011*). In addition, neurons in the PL cortex have been shown to track reward value and reflect impulsive choices (*Sackett et al., 2019*). More generally, the inactivation of PL/IL cortices using optogenetics leads to an increase in premature responses in a probabilistic reversal task (*Nakayama et al., 2018*), while the optogenetic activation of the PL/IL cortices increases food-seeking behavior while reducing impulsive actions (*Warthen et al., 2016*).

Furthermore, lesions in the AC impair behavioral inhibition producing an increase in premature actions in rodents (*Muir et al., 1996*; *Hvoslef-Eide et al., 2018*). More recently, it has been shown that the control of impulsive actions exerted by the AC requires intact signaling through Gi-protein in its layer 5 pyramidal neurons (*van der Veen et al., 2021*). Altogether, there is considerable evidence linking the activity of the areas composing the mPFC (PL, IL, and AC cortices) to the control of impulsive actions.

In addition, the optogenetic activation of DRN 5-HT neurons, a major subcortical target of mPFC projections (*Geddes et al., 2016*; *Zhou et al., 2017*; *Weissbourd et al., 2014*; *Pollak Dorocic et al., 2014*), improves the performance of a delayed response task (*Miyazaki et al., 2014*; *Miyazaki et al., 2018*; *Fonseca et al., 2015*) through an increase in active behavioral persistence (*Lottem et al., 2018*), which is the converse effect of the pharmacological silencing of the mPFC (*Narayanan et al., 2006*; *Narayanan et al., 2013*; *Murakami et al., 2017*). Altogether, the emerging picture suggests that the individual activation of either mPFC or DRN converges into a behaviorally persistent phenotype. Consistent with this, the activation of mPFC–DRN top–down projections also has been shown to increase active persistence (*Warden et al., 2012*). However, previous studies have reported a net inhibitory effect of mPFC input onto 5-HT neurons in the DRN (*Celada et al., 2001*; *Maier, 2015*), particularly after prolonged trains of high-frequency stimulation (*Srejic et al., 2015*). This raises a question on the directionality with which mPFC input modulates DRN neuronal activity in the context of behavioral control. One possible mechanism would be a frequency dependency of the net effect, as found in thalamocortical connections (*Crandall et al., 2015*). In this scenario, given that the inhibitory interneurons in the DRN can track faster frequencies than 5-HT neurons (*Jin et al., 2015*) and that 5-HT neurons undergo 5-HT1a autoreceptor-mediated inhibition upon dendritic NMDA receptor activation (*de Kock et al., 2006*), a prolonged activation of mPFC afferents to the DRN may, in turn, produce inhibition of 5-HT neurons. Nevertheless, other, less explored, patterns of cortical activity in different frequency ranges may tune 5-HT neuron subpopulations in different ways under more naturalistic patterns of activation and could be the focus of future research. An alternative mechanism for the bidirectional control of DRN activity by mPFC input would be synaptic plasticity, since it has been shown that activity-dependent plasticity (*Challis and Berton, 2015*) and neuromodulators (*Geddes et al., 2016*) can bias the net excitatory or inhibitory effect that mPFC input exerts on DRN 5-HT neurons.

In addition, while it is well-described that the PL/IL cortices produce a dense innervation over the DRN, the adjacent PR and IL cortices exert opposite effects on fear conditioning (*Giustino and Maren, 2015*) as well as on avoidance behaviors and behavioral inhibition (*Capuzzo and Floresco, 2020*). This striking contraposition in their functional role leaves open the possibility of different circuit motifs in their DRN innervation that could explain a putative excitatory or inhibitory effect and this should also be the focus of future research.

The cortical subpopulation of DRN-projecting neurons manipulated in adult Rbp4-Cre mice in this study presented collateral projections that were particularly dense onto the dorsomedial striatum (*Figure 3—figure supplement 3*), a pathway that has been shown relevant for foraging decisions (*Bari et al., 2019*). Although it has been shown that the cortico-striatal pathway is fully developed after P14 using Rbp4-Cre mice (*Peixoto et al., 2016*) and therefore unlikely to underlie the developmental differences observed in this study, we cannot rule out an impact of the ablation of cortico-striatal collaterals in the behavioral persistence decrease observed in Caspase-treated mice.

The presence of parallel subsystems in the DRN, with complementary projections either to the prefrontal cortex or to the amygdala and responsible for different behavioral responses has recently been reported (*Ren et al., 2018*). In our hands, cortico-DRN descending neurons had very sparse collateralization to the amygdala (*Figure 3—figure supplement 3*), while collaterals to the dorsal striatum or substantia nigra were abundant. This may suggest the presence of loops of preferential interconnectivity (mPFC → DRN/DRN → mPFC and mPFC → Amygdala/Amygdala → mPFC) as it has been shown for other cortical–subcortical loops (*Young et al., 2021*; *Li et al., 2020*), with different DRN subpopulations exerting specific neuromodulatory effects in either region (*Ren et al., 2018*).

To summarize, our results describe a process of late postnatal development of top–down mPFC afferents onto DRN causally linked to the emergence of behavioral persistence in the transition between adolescence and adulthood. This critical period of corticofugal axonal development may also represent a period of vulnerability for maladaptive development involved in the etiopathogenesis of psychiatric disorders (*Rutter et al., 2007*; *Chen et al., 2018*; *Soiza-Reilly et al., 2019*; *Guirado et al., 2020*).

## Methods

### Animals

All experimental procedures were approved and performed in accordance with the Champalimaud Centre for the Unknown Ethics Committee guidelines and by the Portuguese Veterinary General Board (Direcção-Geral de Veterinária, approval 0421/000/000/2016). The mouse lines used in this study were obtained from the Mutant Mouse Resource and Research Center (MMRRC), Rbp4-Cre (stock number 031125-UCD), and from Jax Mice, Ai32(RCL-ChR2(H134R)/EYFP) (Stock number 012569) and Ai9(RCL-tdTomato) (7905). All of them were backcrossed to C57BL/6 in-house for at least 10 generations prior to their use in our experiments. Mice were kept under a standard 12 hr light/dark cycle with food and water ad libitum. Behavioral testing occurred during the light period.

### Electrophysiological recordings

Male and female mice were used for whole-cell recordings. Coronal slices of 300 µm thickness containing the dorsal raphe were cut using a vibratome (Leica VT1200) in 'ice cold' solution containing (in mM): 2.5 KCl, 1.25 NaH$_2$PO$_4$, 26 NaHCO$_3$, 10 D-glucose, 230 Sucrose, 0.5 CaCl$_2$, 10 MgSO$_4$, and bubbled with 5% CO$_2$ and 95% O$_2$. Slices were recovered in artificial cerebrospinal fluid (ACSF) containing (in mM): 127 NaCl, 2.5 KCl, 25 NaHCO$_3$, 1.25 NaH$_2$PO$_4$, 25 Glucose, 2 CaCl$_2$, 1 MgCl$_2$ at 34°C for 30 min and then kept in the same solution at room temperature until transferred to the recording chamber. In addition, 300 µM L-tryptophan (Sigma) was added to the ACSF to maintain serotonergic tone in the ex vivo preparation as described elsewhere (*Liu et al., 2005*).

Patch recording pipettes (resistance 3–5 MΩ) were filled with internal solution containing (in mM): 135 K-Gluconate, 10 HEPES (4-(2-hydroxyethyl)-1-piperazineethanesulfonic acid), 10 Na-Phosphocreatine, 3 Na-L-Ascorbate, 4 MgCl$_2$, 4 Na$_2$-ATP, and 0.4 Na-GTP. Data were acquired using a Multiclamp 700B amplifier and digitized at 10 kHz with a Digidata 1440a digitizer (both from Molecular Devices). Data were then either analyzed using Clampfit 10.7 Software (Molecular Devices, LLC) or imported into Matlab and analyzed with custom-written software.

To ensure the same recording quality across experiments, access and series resistance was calculated for every cell recorded in voltage-clamp mode using the test pulse mode of Clampex (100 ms, −10 mV). Briefly, access resistance (Ra) was determined by measuring the amplitude of the current response to the command voltage step and the membrane resistance (Rm) as the difference between the baseline and the holding current in the steady state after the capacitive decay, by applying Ohm's law. Input resistance was the sum of the membrane resistance with the pipette resistance. The membrane time constant (Tau, $\tau$) was determined by a single exponential fit of the decay phase in response to the square pulse. An approximation of the capacitance was obtained offline using the following formula:

$$\text{Tau} = \text{Access Resistance} * \text{Input Capacitance}$$

Only neurons with access resistance 1/10th lower than the membrane resistance were used for analysis. Only neurons with an access resistance lower than 30 MOhm were considered for analysis. The access resistance was comparable between neurons recorded at different developmental stages (ANOVA, $F(3,119) = 1.78$, p = 0.15).

Neurons were recorded at a holding membrane voltage of −70 mV, near the reversal potential of chloride (−68 mV) and thus, optogenetically evoked responses correspond to AMPA-mediated currents.

To activate ChR2-expressing fibers, light from a 473 nm fiber-coupled laser (PSU-H-FDA, CNI Laser) was delivered at approximately 2 mm from the sample to produce wide-field illumination of the recorded cell (*Figure 2A*). TTL triggered pulses of light (10 ms duration; 10 mW measured at the fiber tip) were delivered at the recording site with 10 s of intersweep interval. In >90% of the neurons considered in the current study, the stimulus consisted in a single pulse of light per sweep. In the remaining subset of recorded neurons the stimulus consisted of a train of light pulses, of same length and amplitude, delivered at frequencies ranging from 2 to 10 Hz. In this subset of recordings, only the amplitude in response to the first peak of the stimulus train was considered for amplitude analysis. Importantly, no sign of intersweep opsin desensitization (decrease of light-evoked EPSC amplitude with consecutive sweeps) was observed in either type of recordings (data not shown). Every data point represented in *Figure 1B, C* corresponds to the average of 6–10 sweeps per recording.

Neurons were recorded in the dorsomedial and lateral wings portions of the DRN, in two consecutive coronal slices per mouse between bregma levels ~4.3 and ~4.8 (*Figure 1—figure supplement 1*).

To assess ChR2-evoked photocurrent in layer 5 somas, mPFC slices were obtained using the same slicing procedure. 1 µM TTX was added to the bath to prevent escaped spikes in the voltage clamp recordings upon light activation.

## Histology

Mice were deeply anesthetized with pentobarbital (Eutasil) and perfused transcardially with 4% paraformaldehyde (P6148, Sigma-Aldrich). The brain was removed from the skull, stored in 4% paraformaldehyde for 2 hr before being transferred to cryoprotectant solution (30% sucrose in phosphate-buffered saline [PBS]) until they sank. Coronal sections (50 µm) were cut with a freezing sliding microtome (SM2000, Leica).

For axonal quantification in Rbp4-ChR2 mice, we performed anti-GFP immunostaining to enhance the intrinsic signal of the ChR2-fused EYFP reporter. We incubated overnight with the anti-GFP primary antibody at 4 degrees (1:1000, A-6455 Invitrogen, 0.1 M PBS 0.3% tx100, 3% normal goat serum (NGS)). After abundant PBS washes, we incubated the secondary biotinylated anti-Rabbit antibody (711-065-152, Jackson IRL) for 2–4 hr in the same incubation solution at room temperature and finally, after PBS washes, slices were incubated in Alexa488 Streptavidin for 2–4 hr in the same incubation solution at room temperature (S32354, Invitrogen). After final PBS washes, slices were mounted and covered with FluoroGel mounting medium (17985-10, Electron Microscopy Sciences) for posterior imaging.

For NeuN detection, we incubated histological slices (from mice injected with rAAV-tdTomato or rAAV-Caspase) overnight with the anti-NeuN primary antibody at 4 degrees (1:1000, EPR12763 Abcam, 0.1 M PBS 0.3% tx100, 3% NGS). After three PBS washes of 10 min each, we incubated the secondary anti-Rabbit-Alexa488 antibody (A32731, Invitrogen) for 2–4 hr in the same incubation

solution at room temperature and finally, after PBS washes, slices were mounted and covered with FluoroGel mounting medium (17985-10, Electron Microscopy Sciences).

## Image acquisition and analysis

Histological sections were imaged with a Zeiss LSM 710 confocal laser scanning microscope using ×10 and ×25 magnification objectives.

To quantify the axonal density, images containing the DRN or the nucleus accumbens were background subtracted and binarized using constant thresholds in Fiji. After thresholding, binary images were imported into Matlab and analyzed with custom-written software. Images were sampled every 100 µm across the Y axis. The intersections between binary axons and these sampling lines across the Y axis were counted and averaged in bins of 100 µm to estimate axonal density.

Quantification of layer 5 soma densities was obtained in histological slices of Rbp4-tdTomato mice. Confocal images were imported into matlab, and fluorescent somas were detected using the image analysis toolbox of Matlab inside a defined region of interest containing the mPFC (PL/IL). The number of somas was then divided by the area of the region of interest (ROI) to obtain the density of neurons.

Quantification of NeuN expressing neurons was performed using the same protocol used in Rbp4-tdTomato mice. First, we visually inspected the expression pattern of tdTomato expressing neurons after injecting rAAV-tdTomato in the DRN of control mice. We found five cortical areas consistently expressing layer 5 tdTomato+ neurons in at least 5/8 control mice: PR/IL, AC, MO, TeA, Rs. We then acquired confocal images of these five areas in mice injected with rAAV-tdTomato or rAAV-Caspase for analysis. Areas showing expression in one to three mice were not included for analysis. These confocal stacks contained the green fluorescent signal of NeuN detection and the red intrinsic fluorescence of td-Tomato. All confocal images consisted of 10 images stacked in the Z plane, with 3 µm spacing, and that were max projected for analysis. Stacks from PR/IL, AC, and MO were 800 × 600 µm (cortical depth × width) and from TeA, Rs, and M1 were 1400 × 600 µm to adjust to their intrinsically different cortical thickness (*Figure 3—figure supplement 2*). For each brain area/mouse, bilateral stacks were acquired at Bregma levels: PR/IL: 1.5 mm, AC and M1: 1.1 mm, MO: 2.3 mm, TeA and Rs: −3.1 mm. Using custom made software based on Matlab's image analysis toolbox, NeuN somas were detected and their densities binned in depth and averaged across mice for final representation.

## Stereotaxic surgeries and virus injection

Adult mice between 8 and 9 weeks of age were anesthetized with isoflurane (2% induction and 0.5–1% for maintenance) and placed in a motorized computer-controlled Stoelting stereotaxic instrument with mouse brain atlas integration and real-time surgery probe visualization in the atlas space (Neurostar, Sindelfingen, Germany). Antibiotic (Enrofloxacin, 2.5–5 mg/kg, S.C.), pain killer (Buprenorphine, 0.1 mg/kg, S.C.), and local anesthesia over the scalp (0.2 ml, 2% Lidocaine, S.C.) were administered before incising the scalp. Virus injection (experiment group: AAV2retro-flex-EF1A-taCasp3-TEVp; control group: AAV2retro-flex-hSyn-tdTomato) was targeted to DRN at the following coordinates: −4.7 mm AP, 0.0 mm ML, and 3.1 mm DV. The vertical stereotaxic arm was tilted 32 degrees caudally to reach the target avoiding Superior sagittal sinus and Transverse sinuses. Target coordinates were adjusted as follows: −6.64 mm AP, 0.0 mm ML, and −4.02 mm DV. To infect a larger volume of the DRN with the virus, we performed six injections of 0.2 µl using two entry points along the AP axis (−6.54 and − −6.74) and three depths along the DV axis (−4.02, −3.92, and −3.82). The incision was then closed using tissue adhesive (VETBOND™/MC, 3 M, No. 1469 SB). Mice were monitored until recovery from the surgery and returned to their homecages, where they were housed individually. Behavioral testing started at least 1 week after surgery to allow for recovery.

## Behavioral testing

The behavioral box consisted of 1 back wall (16 × 219 cm), 2 side walls (16.7 × 219 cm), and 2 front walls (10 × 219 cm, 140-degrees angle between them), made of white acrylic (0.5 cm thick) and a transparent acrylic lead. A camera (ELP camera, ELP-USBFHD01M-L180) was mounted on top of the ceiling for monitoring purposes. Each front wall had a nose-poke port equipped with an infrared emitter/sensor pairs to report port entry and exit times (model 007120.0002, Island motion corporation) and a water valve for water delivery (LHDA1233115H, The Lee Company, Westbrook, CT). An internal white acrylic wall (8 cm) separates the two nose-poke ports forcing the animals to walk around

it to travel between ports. All signals from sensors were processed by Arduino Mega 2560 microcontroller board (Arduino, Somerville, US), and outputs from the Arduino Mega 2560 microcontroller board were implemented to control water delivery in drops of 4 µl. Arduino Mega 2560 microcontroller was connected to the sensors and controllers through an Arduino Mega 2560 adaptor board developed by the Champalimaud Foundation Scientific Hardware Platform.

Subjects have to probe two foraging sites (nose-poke ports, for mice, or virtual magic wands, for humans) to obtain rewards (4 µl water drops, for mice, or virtual points for humans). At any given time, only one of the sites is active and, when probed, delivers a reward with a fixed 90% probability ($p_{REW}$). Each attempt also triggers a fixed 90% probability of transition ($p_{TRS}$) to inactivate the current foraging site and activate the other. These transitions are not cued; thus, subjects are required to alternate probing the current site and traveling to the other to track the hidden active state and obtain rewards. In this work, we focus on assessing differences in the baseline patience/impulsivity, measured as the ability to withhold adverse outcomes. Therefore naive subjects were only tested once.

Five days before testing, water dispensers were removed from the animals' home cages, and their weights were recorded. In the following days, progressively less water (1000, 800, and 600 µl) was given in a metal dish inside the homecage. Weight loss was monitored every day before water delivery, and no animal lost more than 20% of their body weight. On the fifth day of water deprivation, animals were weighed and introduced to the behavioral box. A small quantity of water was present at the start of the session to stimulate the mice to probe the nose-ports. Sessions lasted a minimum of 1 hr. By that time, if animals did not perform at least 30 trials, the session was extended for thirty more minutes.

Animals were handled during water deprivation to reduce stress levels, but they were completely naive about the task environment and functioning on the testing day. One juvenile female mouse was excluded from the experiment batch before the task assessment because of congenital blindness. One caspase adult mouse was excluded after the task assessment because of abnormal behavior. Rather than nose poking to seek water, this animal spent most of the task time biting the nose-port, to anomalous levels. In chronological order, we tested a batch of only male juvenile and adult animals, followed by testing of male and female tdTomato and Caspase animals, and finally only female juvenile and adult animals. Separate analysis for females and males on the effect of age reveals that juveniles are less persistent in both cases.

## Data and statistical analysis

Behavioral data analysis was performed using custom-written scripts in Julia-1.4.1.

Behavioral results were represented as median ± 95% CIs, and statistical significance was accepted for p-values <0.05. The statistical analysis was done in Julia-1.4.1 (*Bezanson et al., 2017*) with the HypothesisTests (https://juliastats.org/HypothesisTests.jl/v0.9/) and MixedModels (*Bates et al., 2021*) existing packages. The effect of a specific factor on the probability of leaving was tested by applying logistic regression on a generalized linear mixed-effects model, using a Bernoulli distribution for the dependent variable and a Logit link function. For each foraging nose poke, we assigned a boolean label according to whether the animal left the patch after that poke (True) or not (False). We then use logistic regression to explain this leaving choice for each poke according to the elapsed time in the trial (PokeTime), the elapsed trials in the session (Trial), the animal group (Age or Virus), and their interactions. This statistical approach allows us to examine the question of behavioral persistence in terms of probability of leaving after each single poke, expanding the amount of usable data, per animal and counterbalancing the limitation of studying the phenomenon in naive animals exposed to a single session. Furthermore, this technique can test for both additive and multiplicative effects of the factors contributing to behavioral persistence. The individual variability was accounted for through generalized linear mixed models with random intercept and slopes for each mouse (see Methods for the implementation). Before testing we checked for co-linearity between the continuous predictors and confirmed that there was no correlation between the time of poking (Poke Time) and trials elapsed from the beginning of the session (Trial) (PokeTime ~ 1 + Trial + (1 + Trial|MouseID): p = 0.99, *Figure 2—figure supplement 1A*). First, to assess the significance of the estimated coefficients, we calculated their 95% CI by performing a parametric bootstrap of 1000 samples. Only factors whose CI did not include 0 were considered to be significantly affecting the probability of leaving. Next, to validate the relevance of the experimental manipulation (age or virus), we compared nested models (a

general model and a special case model, excluding or including the experimental factor, respectively) using a likelihood ratio test: Chi-squared test on the difference of the deviance of the two nested models, with degrees of freedom equal to the difference in degrees of freedom between the general model (lacking the predictor) and its special case (with the predictor of interest). For each analysis, we report the median and 95% CI of the median for the groups of interest, followed by the test statistics. We use Wilkinson annotation to describe the models with denoting random effects.

Electrophysiological and histological results were analyzed with Matlab and GraphPad Software. Normality of the residuals was tested with the D'Agostino–Pearson omnibus K2 test. When normally distributed, either a $t$-test, one-way ANOVA or repeated measures ANOVA were performed to compare groups at different developmental phases. In the cases where residuals were not normally distributed, we performed a Mann–Whitney or Kruskal–Wallis test to assess significance. For testing differences in connection probability, a Chi-square test was performed. Finally, a Kolmogorov–Smirnoff test was performed to compare the neuronal density distribution between Caspase-treated animals and tdTomato expressing controls. Error bar plots represent mean ± SEM. Significance was noted as *p < 0.05.

## Acknowledgements

We thank Drs. Cindy Poo and Constanze Lenschow for helpful comments on the manuscript and the Champalimaud Foundation Advanced Bio-optics and Bio-imaging platform for the microscopy technical assistance. This work was supported by the Champalimaud Foundation (ZFM), European Research Council (671251, ZFM), and Fundação para a Ciência e Tecnologia (FCT-PTDC/MED-NEU/28830/2017, ZFM; SFRH/BD/132172/2017, DS). This work was further supported by Portuguese national funds Fundação para a Ciência e a Tecnologia (FCT; UIDB/04443/2020); CONGENTO, co-financed by Lisboa Regional Operational Programme (Lisboa2020), under the PORTUGAL 2020 Partnership Agreement, through the European Regional Development Fund (ERDF) and Fundação para a Ciência e Tecnologia (Portugal) under the project LISBOA-01-0145-FEDER-022170, the imaging platform has been financed under the project LISBOA-01-0145-FEDER-022122.

## Additional information

### Funding

| Funder | Grant reference number | Author |
| --- | --- | --- |
| European Research Council | 671251 | Zachary F Mainen |
| Fundação para a Ciência e a Tecnologia | FCT-PTDC/MED-NEU/28830/2017 | Zachary F Mainen |
| Fundação para a Ciência e Tecnologia | SFRH/BD/132172/2017 | Dario Sarra |
| Champalimaud Foundation | | Zachary F Mainen |

The funders had no role in study design, data collection, and interpretation, or the decision to submit the work for publication.

### Author contributions

Nicolas Gutierrez-Castellanos, Conceptualization, Data curation, Formal analysis, Investigation, Visualization, Writing – original draft; Dario Sarra, Conceptualization, Data curation, Formal analysis, Investigation, Visualization, Writing – original draft, Funding acquisition, Methodology, Writing – review and editing; Beatriz S Godinho, Investigation; Zachary F Mainen, Conceptualization, Supervision, Funding acquisition, Writing – review and editing

### Author ORCIDs

Nicolas Gutierrez-Castellanos  http://orcid.org/0000-0002-8442-4243
Dario Sarra  http://orcid.org/0000-0002-6800-8071
Zachary F Mainen  http://orcid.org/0000-0001-7913-9109

## Ethics

All experimental procedures were approved and performed in accordance with the Champalimaud Centre for the Unknown Ethics Committee guidelines and by the Portuguese Veterinary General Board (Direcèäo-Geral de Veterinâria, approval 0421/000/000/2016).

## Decision letter and Author response

Decision letter https://doi.org/10.7554/eLife.93485.sa1
Author response https://doi.org/10.7554/eLife.93485.sa2

## Additional files

### Supplementary files

• MDAR checklist

### Data availability

All data analyzed and visualized during this study are included in form of Source Data files that have been provided for all figures present in the current manuscript.

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
