## [Editor Report]

In this important study, the authors explore the importance of developmental changes in cortico-DRN innervation in the balance of behavioral control in a foraging task. The authors report somewhat convincing evidence that while juvenile mice and adult mice both perform the task, juveniles exhibit more impulsive behavior due to reduced efficacy of cortico-DRN projections. The authors conclude that the development of cortico-DRN projections allows 5HT input to promote perseveration (or exploitation) in the balance of behavioral control.

---

## [Decision Letter]

**Decision letter after peer review:**

[Editors’ note: the authors submitted for reconsideration following the decision after peer review. What follows is the decision letter after the first round of review.]

Thank you for submitting the paper "Maturation of prefrontal input to dorsal raphe nucleus increases behavioral persistence in mice" for consideration by *eLife*. Your article has been reviewed by 2 peer reviewers, including Geoffrey Schoenbaum as the Reviewing Editor and Reviewer #1, and the evaluation has been overseen by a Senior Editor.

Comments to the Authors:

We are sorry to say that, after consultation with the reviewers, we have decided that this work will not be considered further for publication by *eLife*. Both reviewers noted the strengths of the work, but both also thought that there were some weaknesses in the claim that the behavioral changes hinged on development of the mPFC-DRN projections and also in the behavioral effect. While it was felt these could be dealt with by modifying the text and framing, there was not a consensus that doing this would still allow meaningful conclusions to be drawn. So it was ultimately felt that addressing the concerns likely would require significant work beyond the scope of a revision.

*Reviewer #1 (Recommendations for the authors):*

This study has a lot to recommend it. It is founded on the idea that 5HT promotes waiting, and tests a clear and I think novel hypothesis that input from prefrontal areas is key to promoting this, and that the increase in this relates to declines in impulsive behavior during adolescence. It also nicely tests that hypothesis with integrated behavioral, electrophysiological and tracing approaches. Overall it makes a compelling argument in favor of the authors results. The independent findings also build upon or at least are well supported by prior work, which I think is excellent and increases confidence in the conclusions.

However I think there are two main caveats or issues to address. The first is that there is a strong focus throughout on the development of mPFC projections to DRN. Yet the key causal manipulation is not specific for this input it seems to me, but instead will target equally input from anywhere really. The second issue is the behavioral evidence of major foraging or waiting problems is relatively weak. The juvenile and adult mice both did the task, and it looks like they did it similarly? Obtaining similar amounts of reward? The differences they show look meaningful and reliable, but they are in details of how they interact with the nosepoke and apparatus it seems like to me. And that they do not appear to translate to clear reductions in the ability to do the task seems to call into some question the dependence of the task on the control impulsivity regulated by the proposed DRN circuit.

As noted, I think there are two main caveats or issues to address. The first is that there is a strong focus throughout on the development of mPFC projections to DRN. Yet the key causal manipulation is not specific for this input it seems to me, but instead will target equally input from anywhere really. I believe the authors will argue still that mPFC areas had the highest density of such projections in their data, however I think this does not rule out that their behavioral effect could be partly or even exclusively due to effects on projections from elsewhere. At a minimum this should be noted in the discussion and perhaps their arguments in favor of this specific circuit made more succinctly so readers can make their own judgement.

The second issue is the behavioral evidence of major foraging or waiting problems is relatively weak. The juvenile and adult mice both did the task, and it looks like they did it similarly? Obtaining similar amounts of reward? The differences they show look meaningful and reliable, but they are in details of how they interact with the nosepoke and apparatus it seems like to me. And that they do not appear to translate to clear reductions in the ability to do the task seems to call into some question the dependence of the task on the control impulsivity regulated by the proposed DRN circuit. Possibly this would be addressed by a more straightforward analysis of the behavioral effects, similar to prior studies, or the authors should make more clear why the modest changes in leaving and port occupancy measures are related to foraging per se.

*Reviewer #2 (Recommendations for the authors):*

This manuscript seeks to answer an important question about how behavioral persistence changes from adolescence to adulthood, and whether medial prefrontal top-down control of the dorsal raphe nucleus (DRN) is responsible for the observed behavioral changes in mice. Using ChR2-assisted circuit mapping, the authors show that cortical layer 5 afferents to the DRN increase in connection probability and potency during adolescence. However, based on the genetic approach used, these findings are not anatomically specific to afferents originating in medial prefrontal cortex (mPFC). Furthermore, the possibility that connection differences were attributable to the number of Rbp4-cre cells at different ages rather than growth of connections from a stable population was not excluded because the authors did not identify and quantify all regions from which these connections are received. This makes the relevance of anatomical findings to the mPFC-DRN pathway unclear.

Performance in a self-paced, operant foraging task was then compared in adolescent and adult mice. Mice in the two age groups appeared to perform differently in the task, with juveniles leaving response locations sooner than adults. However, based on the underlying statistical properties of the task, the more frequent location switching displayed by juveniles may have been a more efficient strategy, compared to adults persisting in the same location for too long. More relevant behavioral measures, such as mean consecutive failed responses and rewards earned per response are needed to fully evaluate task performance. Evidence of bimodality in the behavioral data should also be noted, as it suggests that not all subjects within a given group used consistent strategies or performed similarly.

Finally, the authors sought to assess changes in task performance when this mPFC-DRN pathway was disrupted in adult mice. To address this, they ablated layer 5 cortical neurons that projected to the DRN. In some measures, ablated adults performed more like juveniles in the same foraging task. However, the ablation technique used was like the anatomical analysis not specific to projection neurons from the mPFC to the DRN. In addition, the retrograde vector used in this experiment labeled collateral projections from cortical neurons to the striatum and other regions, making the scope of potential substrates quite large. Therefore, the behavioral outcomes were likely driven at least in part by the loss of layer 5 projection neurons from other cortical areas, or from the loss of their collaterals to areas outside of the DRN.

1. The reader would benefit from more justification/background on why using Rbp4-cre mice is appropriate for addressing the research question here. Furthermore, more information about the proportion of layer 5 neurons that successfully express ChR2 using this method should be described beyond "a large fraction" (Pg 6, line 112).

2. If Rbp4-Cre/ChR2-loxP mice were used to assess the effects of laser evoked firing in the DRN, we can't be sure the excised axons originated from the mPFC. Since these data reflect changes in general cortical input to DRN, and the control experiments to rule out differences in ChR2 expression were only done in mPFC, these limitations should be addressed. The authors should either use more selective techniques and include more specific analysis of the mPFC-DRN pathway or eliminate these specific conclusions and reserve them for speculation.

3. The description of the behavioral task is confusing and seems to obscure the fact that (based on the high probability of reward and high probability of switch used-0.9 each) the optimal strategy is to be flexible in response location, rather than persistent. More details are needed on the task: what determines trial start? What is the optimal strategy to use? What is the average number of responses before a subject should switch? Why do the authors state that this task "requires persistence in poking at the port despite reward failures" when it appears subjects are meant to be switching response locations frequently?

4. The text describes that each response in the active port carries a 0.9 probability that the active site will switch to the other location. This would seem to encourage alternation behavior after about 2 responses, not persistent behavior in one location. Are these data, therefore, reflecting maladaptive perseveration in adults, rather than adaptive persistence?

5. There are key behavioral outcome measures missing from the analysis that would make it much stronger. Mean number of rewards earned per response should be provided to give the reader more information about how efficiently the juvenile vs. adult strategies performed.

6. Some background on what is known about juvenile cognition would help put the results in context and appreciate any potential confounds. For example, juveniles are more impulsive.

7. In addition, the Vertechi et al., (2020) paper referenced for this behavioral task states that, "We found that the number of consecutive failures since the last reward (ConsecutiveFailureIndex) was a better predictor of mouse choice than the time spent at the nose poke". Number of consecutive failures should be reported here. Including this more informative measure is important especially when the time-based measures used here were vulnerable to task-irrelevant exploratory behavior, which skewed foraging episode times, and required the data the be reanalyzed with an arbitrary cutoff time of 60s.

8. The measure of time spent poking does not seem informative, or at least it does not give the reader more information than the number of pokes produced in a given bout.

9. The ablated pathway was not mPFC-DRN specific, but instead targeted any Rbp4-expressing neurons projecting to the DRN. The authors concluded that the highest density of DRN projections originated in mPFC, but only provided data from a small number of cortical comparison regions (temporal association cortex, retrosplenial cortex, and M1). Furthermore, somas expressing the control rAAV were shown to send collaterals to several regions outside of the DRN, such as the VTA, PAG, and the striatum. The loss of these collaterals in the ablation group could impact behavioral outcomes. Therefore, the data presented here are not sufficient to conclude that observed behavioral effects were specifically driven by mPFC-DRN pathway ablation. The language attributing behavioral changes exclusively to mPFC-DRN pathway ablation should be changed to reflect this lack of anatomical specificity.

10. The quantification of NeuN-expressing cell density was not described in the methods or main text. Since NeuN-labeling was used to quantify/confirm neuronal density loss in ablation mice compared to controls, the quantification process should be described fully in the methods.

11. The kernel density functions for 'leaving time' appear bimodal in several cases, especially in Figure 2C. The possibility that individuals within the same group/condition are using different strategies or are displaying different behavioral pattens should be addressed.

[Editors’ note: further revisions were suggested prior to acceptance, as described below.]

Thank you for submitting the paper "Maturation of cortical input to dorsal raphe nucleus increases behavioral persistence in mice" for consideration by *eLife*. Your article has been reviewed by 3 peer reviewers, one of whom is a member of our Board of Reviewing Editors, and the evaluation has been overseen by a Senior Editor.

Comments to the Authors:

While we appreciated your responses to the original reviews, because of the very long time between the initial submission and this revision, some of the prior reviewers were not available. As a result, it was necessary to obtain new reviewers, who raised additional substantial concerns regarding some of the methodology. They felt that these issues required substantial additional control work.

*Reviewer #2 (Recommendations for the authors):*

This study characterizes the development of cortico-raphe projections from L5 cortical neurons and their influence on foraging and persistent behavior. The finding that cortico-raphe projections mature late in development is extremely interesting and raises several new questions regarding the role of these projections in adolescence. Most of the significant points have already been addressed by other reviewers; I will focus on a few technical aspects of the electrophysiology experiments and the caspase3-based approach.

The late development or cortico-raphe projections is well supported by the histological and optogenetic experiments depicted in Figure 1. However, the reported oEPSC values seem to be based on a single trial/stimulus, is that correct? The authors mention "TTL-triggered pulses of light (10-ms duration; 10 mW measured at the fiber tip) were delivered at the recording site with a 10-second inter-sweep interval. In cases where multiple pulses were delivered per sweep, only the first one was considered for analysis to eliminate short-term plasticity effects on the measured amplitude." – A 10-second inter-sweep interval is insufficient to recover opsin desensitization, potentially leading to the perceived short-term plasticity. Increasing the inter-sweep interval to >45 seconds should resolve this issue and enable the averaging of individual sweeps to minimize the variability inherent in opto-evoked whole-cell recordings.

The control experiments based on NeuN IHC do not provide an estimate of the fraction of cortico-raphe projections ablated by the retro caspase approach, which is an important aspect to better interpret the behavioral results. This is because AAVrg only infects a subset of neurons. There are three potential alternative approaches to characterize the extent of cortico-raphe projection loss. One is to perform a second AAVrg-tdTom injection two weeks after AAVrg-DIO-casp3 to compare how many cells can still be retro-infected after the first caspase injection. However, quantifying these experiments is not straightforward, and variability due to viral injection efficiency and targeting is always a confounding factor. A more rigorous and cleaner approach could be to inject AAVrg-DIO-casp3 in Rbp4-Cre x Ai32 mice and compare oEPSC amplitudes in injected vs non-injected mice. As ChR2 is stably expressed in these mice this would provide a better assessment of connectivity changes after caspase ablations. A third approach would be to use Rbp4-Cre x XFP mice and quantify cortico-raphe fiber density changes after AAVrg-DIO-casp3 (similar to what is shown in Figure 1).

However, these strategies would only probe the fraction of cortical neurons expressing Cre, which, in the case of Rbp4-Cre, is approximately 50% of L5 neurons. It's possible that Rbp4-negative L5 neurons also project to the raphe, and these are not affected by DIO-casp3 ablation. Quantifying the extent of total cortico-raphe ablation is an important point, as considerable remaining fibers could lead to an underestimation of the behavioral effects caused by cortico-raphe manipulations.

Unless I missed it, there is no description of the age at which the caspase injection is performed in the different experiments. This is an important experimental detail.

Regarding the specificity of mPFC>raphe projections raised by other reviewers, the authors discuss the possibility of other mPFC collaterals in the striatum. Future experiments inhibiting local terminals (perhaps using Emx1-Cre crosses) with PPO/eOPN3 or activating ChR2 fibers in juveniles will help further support these findings.

*Reviewer #3 (Recommendations for the authors):*

The paper in general gives very little detail on the anatomical analysis,

For the developmental description, It would be good to show to what extent the Rbp4 labels the PFC-raphe projection. Convincing would be for instance a retrograde labeling from the raphe in the Rbp4-GFP with some quantitative estimate.

For the analysis of afferent axons, the methods authors state that the analysis was done on sagittal sections, in which the position of the DRN and accumbens is not very easy to identify. Yet In figure 1 they show coronal sections.

Additionally, at this resolution and without co-labeling with a synaptic marker they cannot distinguish an increase of fluorescence or passing fibres from a true increase in the number of terminals.

For the retrograde lesion studies, they need to show the injection site of their viral delivery to determine how this could have impacted neighbouring structures. They also could better quantify the specific loss of l5 neurons with an independent L5 marker such as Ctip2.

In all the morphological analysis they need to show where the measures were done. They also need to indicate more precisely how the measures were made (field size analyzed, precise steps of image processing, magnification, number of sections analysed /case).

---

## [Author Response]

[Editors’ note: the authors resubmitted a revised version of the paper for consideration. What follows is the authors’ response to the first round of review.]

Comments to the Authors:We are sorry to say that, after consultation with the reviewers, we have decided that this work will not be considered further for publication by eLife. Both reviewers noted the strengths of the work, but both also thought that there were some weaknesses in the claim that the behavioral changes hinged on development of the mPFC-DRN projections and also in the behavioral effect. While it was felt these could be dealt with by modifying the text and framing, there was not a consensus that doing this would still allow meaningful conclusions to be drawn. So it was ultimately felt that addressing the concerns likely would require significant work beyond the scope of a revision.Reviewer #1 (Recommendations for the authors):This study has a lot to recommend it. It is founded on the idea that 5HT promotes waiting, and tests a clear and I think novel hypothesis that input from prefrontal areas is key to promoting this, and that the increase in this relates to declines in impulsive behavior during adolescence. It also nicely tests that hypothesis with integrated behavioral, electrophysiological and tracing approaches. Overall it makes a compelling argument in favor of the authors results. The independent findings also build upon or at least are well supported by prior work, which I think is excellent and increases confidence in the conclusions.

We are grateful for the reviewer’s positive comments.

However I think there are two main caveats or issues to address. The first is that there is a strong focus throughout on the development of mPFC projections to DRN. Yet the key causal manipulation is not specific for this input it seems to me, but instead will target equally input from anywhere really. The second issue is the behavioral evidence of major foraging or waiting problems is relatively weak. The juvenile and adult mice both did the task, and it looks like they did it similarly? Obtaining similar amounts of reward? The differences they show look meaningful and reliable, but they are in details of how they interact with the nosepoke and apparatus it seems like to me. And that they do not appear to translate to clear reductions in the ability to do the task seems to call into some question the dependence of the task on the control impulsivity regulated by the proposed DRN circuit.As noted, I think there are two main caveats or issues to address. The first is that there is a strong focus throughout on the development of mPFC projections to DRN. Yet the key causal manipulation is not specific for this input it seems to me, but instead will target equally input from anywhere really. I believe the authors will argue still that mPFC areas had the highest density of such projections in their data, however I think this does not rule out that their behavioral effect could be partly or even exclusively due to effects on projections from elsewhere. At a minimum this should be noted in the discussion and perhaps their arguments in favor of this specific circuit made more succinctly so readers can make their own judgement.

We agree with the reviewer’s assessment that our circuit mapping approach is not completely mPFC specific (Rbp4-Cre line). In the original manuscript we referred to cortico-DRN projections until we presented evidence that most projections indeed arise from prefrontal cortical areas (Figure 3A-C and Figure S4). We found that projections from mPFC areas are 13 times more abundant than those originating in other cortical areas, which we believe is reasonable evidence to suggest it is the principal player in the phenotypic differences we described. However, we concur with the reviewer’s comment that we can still not rule out the involvement of the minor DRN afferent projections originating outside of the mPFC (including Rs and TeA cortices). Therefore, to be more conservative in our interpretation, in our revised manuscript, we refer to “cortico-raphe” projections throughout the results (changes are highlighted in bold font):

Title: “Maturation of **cortical** input to dorsal raphe nucleus increases behavioral persistence in mice”

Abstract: Lines 20-28: “Here, we used a genetic approach to describe the maturation of the projection from layer 5 neurons of the neocortex to the dorsal raphe nucleus in mice. Using optogenetic assisted circuit mapping, we show that this projection undergoes a dramatic increase in synaptic potency between postnatal weeks 3 and 8, corresponding to the transition from juvenile to adult. We then show that this period corresponds to an increase in the behavioral persistence that mice exhibit in a foraging task. Finally, we used a genetic targeting strategy that primarily affected neurons in the medial prefrontal cortex (mPFC), to selectively ablate this pathway in adulthood and show that mice revert to a behavioral phenotype similar to juveniles.”

Introduction: Lines 114-122: “First, using a transgenic line (Rbp-Cre) that targets the layer 5 neurons that provide the neocortical input to the DRN, we discovered that this input undergoes a dramatic increase in potency over the course of development from 3-4 weeks (juvenile) to 7-8 weeks (adult). Then, using a probabilistic foraging task, we found that mice’s behavior persistence increased over the same period. Finally, using a genetic ablation technique that primarily affected the mPFC, we showed that ablation of neocortical input to the DRN in adult mice recapitulated the juvenile foraging behavior. Together, these results identify a descending neocortical pathway to the DRN that is critical to the maturation of behavioral control that characterizes adulthood.”

Results: Lines 349-350: “Cortico-DRN pathway ablation in adult mice recapitulates juvenile behavioral features.”

Lines 352-355: “To test the causal link between the development of cortico-raphe afferents and the observed increase in persistence during foraging, we next ablated the cortico-raphe pathway in adult mice and assessed the impact on behavioral persistence.”

Lines 435-439: “Together, these results show that turning off the cortical input to the DRN, which mostly comes from the mPFC, makes adult mice behave like young mice when they are performing the same foraging task. This means mature cortico-DRN innervation is necessary for adult mice to be persistent in their behavior, and this pathway is likely to help mice learn to be persistent in their behavior.”

Discussion: Lines 472-478: “In the present study, we described how the postnatal maturation of the cortical innervation over the DRN during adolescence is linked to the performance of a probabilistic foraging task. Over the same period of development, the cortico-raphe projections underwent a dramatic increase in potency and mice developed an increase in persistence in foraging behavior. Ablation of this pathway in adult mice recapitulated the features observed in the behavior of juvenile mice, supporting a causal relationship between the cortico-raphe input and behavioral persistence.”

Lines 504-505: “However, the specific contribution of long-range top-down cortical circuits and the cellular mechanisms underlying its development had not been previously investigated.“

Lines 513-521: “Thus, our findings are consistent with previous observations in the literature and suggest that the maturation of cortico-DRN afferents starts early in postnatal development and undergoes an extended development period, plateauing only after reaching 7-8 weeks of age. Among the previous studies investigating the postnatal development of top-down afferents from the neocortex in rodents (Klune et al., 2021, Peixoto et al., 2016, Ferguson & Gao 2015), the latest afferent maturational process reported is the mPFC innervation over the basolateral amygdala, which occurs up to week 4 (Arruda-Carvalho et al., 2017). Thus, to our knowledge, the cortical innervation of the DRN represents the latest top-down pathway to develop.”

And leave the role of mPFC-DRN projections as a discussion topic:

Lines 532-535: “Furthermore, we localized the origin of these projections and quantified the local neuronal loss. The PL, IL, and AC cortices, areas that comprise the so called mPFC (Klune et al., 2021), suffered a significant loss with the procedure. Although we cannot rule out a contribution of the other affected areas (Rs and TeA) in our caspase manipulation experiment, it is very likely that the areas with higher neuronal loss (mPFC) made a pivotal contribution to the behavioral changes we observed.”

The second issue is the behavioral evidence of major foraging or waiting problems is relatively weak. The juvenile and adult mice both did the task, and it looks like they did it similarly? Obtaining similar amounts of reward? The differences they show look meaningful and reliable, but they are in details of how they interact with the nosepoke and apparatus it seems like to me. And that they do not appear to translate to clear reductions in the ability to do the task seems to call into some question the dependence of the task on the control impulsivity regulated by the proposed DRN circuit. Possibly this would be addressed by a more straightforward analysis of the behavioral effects, similar to prior studies, or the authors should make more clear why the modest changes in leaving and port occupancy measures are related to foraging per se.

In the present study we used a foraging task to provide a metric of behavioral persistence that was available with minimal training. We took this approach because of the need to measure persistence over the brief course of life of a juvenile animal. The behavior of such juvenile mice have only rarely been studied in any situation. The task was well-suited to this purpose because we found, to our initial surprise, that juvenile mice and adult mice both performed it to some degree on the first day of exposure, unlike any other task for measuring persistence that we are familiar with.

The main point we want to argue is that there are differences in persistence in poking between the two groups. We summarized the differences in foraging behavior between juvenile/adult and caspase/control mice in terms of elapsed time because we found that leaving decisions were better explained as a function of time than pokes. This evidence came from comparing which model (under a constant number of degrees of freedom) fitted better the data using the Akaike information criterion (AIC): Leave~1+PokeTime+(1+PokeTime|MouseID) versus Leave~1+PokeNumber+(1+PokeNumber|MouseID): AICtime=1.13e4, AICnumber=1.14e4 p: 1.52e-23).

However, to address the reviewer's concern, we now include the analysis of pokes after last, as shown in previous reports from our group (data now included in Figure 2 panel F and Figure 3 panel E). This analysis confirms that juvenile mice display reduced persistence compared to controls, and also that the caspase ablation recapitulates the same effect which strengthens the claims we describe to the reader:

Lines 297-309: “Although the time elapsed from the beginning of a trial is a better metric to explain leaving decisions, it does not distinguish between active persistence and spurious pauses in poking. Changes in leaving time could be caused either by an increase in the number of attempts performed or by an increase in the time between attempts. We therefore compared the number of attempts per trial in adults and juveniles, both as overall number of pokes and as consecutive unrewarded pokes performed after the last reward (Vertechi et al., 2020) (Figure 2F). Adults made more pokes per trial than juveniles (Figure 2F; Adults: median = 3.65 pokes per trial, 95% CI = [0.42, 0.51], Juveniles: median = 2.92 pokes per trial, 95% CI = [0.35, 0.50]; Mann-Whitney U test (N Adults = 21, N Juveniles = 23) = 354.5, p = 0.008, effect size = 0.73), and more consecutive failures after the last reward (Figure 2F; Adults: median = 2.69 pokes after last reward, 95% CI = [0.39, 0.53], Juveniles: median = 1.96 pokes after last reward, 95% CI = [0.41, 0.52]; Mann-Whitney U test (N Adults = 21, N Juveniles = 23) = 351.0, p = 0.009, effect size = 0.73).”

**Author response image 1. sa2fig1:** 

While the phenotypic differences we report are relatively small in magnitude, as noted by the reviewer, they were reliable across our experimental groups. Importantly, in a previous study from our group, it was shown that the direct optogenetic stimulation of DRN 5HT neurons produces similarly small and reliable persistence differences in mice performing a probabilistic foraging task (Lottem et al., 2018). Thus, the magnitude of the differences here reported are in line with our predictions for the manipulation of excitatory afferent pathways onto DRN neurons, as we now report in the discussion:Lines 522-530: “Importantly, we found that the structural development of cortico-DRN projections is causally linked to the maturation of behavioral persistence in adult mice. Using a genetically driven ablation approach (Yang et al., 2013), we selectively eliminated layer 5 cortical neurons projecting to DRN in adult mice. The procedure resulted in a behavioral phenotype that replicated key features of the juvenile foraging behavior. We observed a reduction in behavioral persistence.

This difference in behavioral persistence was small but reliable, and of similar magnitude (but, as expected, in the opposite direction) to the difference observed when optogenetically stimulating 5HT DRN neurons in mice performing a probabilistic foraging task (Lottem et al., 2018).”

Finally, despite these differences, we do not argue that the foraging strategy adopted by juvenile mice is worse than that of adult mice or that the cortico-raphe projection is responsible for the ability of mice to perform the task adequately. To the contrary, we highlight the overall similar performance, suggesting that the foraging behavior is relatively “innate” and its basic execution does not solely depend on cortico-raphe activity.

Indeed, as the reviewer suggests, juvenile and adult mice performed the task overall similarly, as illustrated by the fact that they gathered rewards at a similar rate (data now included in Figure 2 panel C and Figure 3 panel E):

Lines 249-259: “We compared the behavior of juvenile (weeks 3-4) and adult mice (weeks 7-8) on their first exposure to the apparatus and task, ensuring that differences in persistence do not arise from differences in learning about the task. We used an environment characterized by high reward probability (Prwd = 90%) and high site-switching probability (Psw = 90%) (Figure 2A). These statistics produce a small number of rewards per trial (Rewards per trial: minimum = 0, maximum = 3) (Figure 2B), and maximize the number of trials performed in one session. Both groups obtained a comparable reward rate (Figure 2C; Adults: median = 0.02 rewards per second, 95% CI = [0.0020, 0.003], Juveniles: median = 0.023 rewards per second, 95% CI = [0.001, 0.012]; Mann-Whitney U test (N Adults = 21, N Juveniles = 23) = 182.0, p = 0.16), indicating that juveniles and adults do not differ in terms of overall competence in performing the task. “

Lines 398-404: “We then assessed the impact of ablation of cortical input to the DRN on behavioral persistence using the same foraging paradigm and analysis we adopted to assess persistence in adults and juveniles (Figure 3D). First, we confirmed that both groups can perform the task comparatively well, obtaining a similar reward rate (Figure 3E; tdTomato: median = 0.033 rewards per second, 95% CI = [0.019, 0.003], Caspase: median = 0.034 rewards per second, 95% CI = [0.010, 0.008]; Mann-Whitney U test (N tdTomato = 8, N Caspase = 7) = 21.0, p = 0.46). ”

Reviewer #2 (Recommendations for the authors):This manuscript seeks to answer an important question about how behavioral persistence changes from adolescence to adulthood, and whether medial prefrontal top-down control of the dorsal raphe nucleus (DRN) is responsible for the observed behavioral changes in mice. Using ChR2-assisted circuit mapping, the authors show that cortical layer 5 afferents to the DRN increase in connection probability and potency during adolescence. However, based on the genetic approach used, these findings are not anatomically specific to afferents originating in medial prefrontal cortex (mPFC). Furthermore, the possibility that connection differences were attributable to the number of Rbp4-cre cells at different ages rather than growth of connections from a stable population was not excluded because the authors did not identify and quantify all regions from which these connections are received. This makes the relevance of anatomical findings to the mPFC-DRN pathway unclear.Performance in a self-paced, operant foraging task was then compared in adolescent and adult mice. Mice in the two age groups appeared to perform differently in the task, with juveniles leaving response locations sooner than adults. However, based on the underlying statistical properties of the task, the more frequent location switching displayed by juveniles may have been a more efficient strategy, compared to adults persisting in the same location for too long. More relevant behavioral measures, such as mean consecutive failed responses and rewards earned per response are needed to fully evaluate task performance. Evidence of bimodality in the behavioral data should also be noted, as it suggests that not all subjects within a given group used consistent strategies or performed similarly.Finally, the authors sought to assess changes in task performance when this mPFC-DRN pathway was disrupted in adult mice. To address this, they ablated layer 5 cortical neurons that projected to the DRN. In some measures, ablated adults performed more like juveniles in the same foraging task. However, the ablation technique used was like the anatomical analysis not specific to projection neurons from the mPFC to the DRN. In addition, the retrograde vector used in this experiment labeled collateral projections from cortical neurons to the striatum and other regions, making the scope of potential substrates quite large. Therefore, the behavioral outcomes were likely driven at least in part by the loss of layer 5 projection neurons from other cortical areas, or from the loss of their collaterals to areas outside of the DRN.1. The reader would benefit from more justification/background on why using Rbp4-cre mice is appropriate for addressing the research question here. Furthermore, more information about the proportion of layer 5 neurons that successfully express ChR2 using this method should be described beyond "a large fraction" (Pg 6, line 112).

We wanted a method that could select the prefrontal input to the DRN. We chose the Rb4-cre line because, as reported by Gerfen and colleagues: “The BAC-Cre driver line Rbp4_KL100 may be considered a pan layer 5 line, displaying expression restricted to most layer 5 neurons throughout neocortical and peri-allocortical areas.” (Gerfen et al., Neuron 2014). Furthermore, Tervo and colleagues have used the Rbp4 line to validate the efficiency of retroAAV vectors, showing that the pattern of cortical projections observed after pontine injections matches that one obtained with standard tracing methods (i.e Fluorogold) and therefore supporting the pan-cortical behavior of the Rbp4 line (Tervo et al., Cell 2016). These references have now been added to the manuscript. While this did not leave us with a perfectly selective labeling of the target pathway, it was in our view the most precise method readily available to us. And, as we observed, mPFC projections are 13.1 times more abundant than those coming from outside the mPFC. Furthermore, other more specific methods based on local virus injections suffer from significant experimental caveats. To assess behavior of juvenile mice at P21-P25, surgeries should be performed at around P10. At this stage, not only virus injection volumes and locations are challenging to match with adults to obtain comparable transfections, but specially, invasive surgical procedures in juvenile rodents have been shown to exert significant early life stress that leads to long lasting behavioral perturbations (Ririe et al., 2021. Front Behav Neurosci).

Therefore we selected an unbiased method that does not depend on injection size or location, that does not suffer from expression artifacts over development (as shown in Figure S1) and that does not represent a source of unwanted early life stress. The current approach allowed us to confirm the behavioral contribution of cortical projections to DRN, and to identify which areas are more likely to be crucial. Future study will benefit from this unbiased approach to guide the use of more selective techniques. A summary of these arguments has now been added to the manuscript:

Results:

Lines 127-136: “First, to characterize the development of neocortical projections to the DRN, we focused on the afferents of layer 5 neurons, which are the primary origin of these projections (Pollak Dorocic, 2014). To do so, we used a mouse line expressing channelrhodopsin-2 (ChR2) under the Rbp4 promoter (Rbp4-Cre/ChR2-loxP) which targets both intra- and extracortical projecting layer 5 neurons (Leone et al., 2015; Gerfen et al., 2013; Tervo et al., 2016) and that has been previously used to map the postnatal development of extracortical projections (Peixoto et al., 2015). Importantly, this approach represents key advantages over alternative viral based strategies as it is insensitive to injection size and location and avoids surgeries in pup mice, thus not introducing an unwanted source of early life stress (Ririe et al., 2021). ”

2. If Rbp4-Cre/ChR2-loxP mice were used to assess the effects of laser evoked firing in the DRN, we can't be sure the excised axons originated from the mPFC. Since these data reflect changes in general cortical input to DRN, and the control experiments to rule out differences in ChR2 expression were only done in mPFC, these limitations should be addressed. The authors should either use more selective techniques and include more specific analysis of the mPFC-DRN pathway or eliminate these specific conclusions and reserve them for speculation.

As pointed out in the response to reviewer 1, we acknowledge that the use of Rbp4 line is not restricted to mPFC. Therefore, in the submitted version we presented these results as “maturation of cortico-DRN input” until providing evidence that most input is, in fact, originating in the mPFC. When injecting, in the DRN of Rbp4 mice, retroAAV viral vectors expressing td-Tomato in a cre dependent manner, we reliably observe retrogradely labeled neurons in 5 cortical areas (in more than 5 out of 8 mice): MO, PL/IL, AC, TeA and Rs. In a recent study mapping whole-brain inputs to the DRN (Xu et al., *ELife* 2021), 8 major areas of projection are identified from the neocortex, 5 of which overlap with our observations. Out of the areas in which we observed projections, not only the majority of them were part of the mPFC, but also the density of these mPFC projections was much higher than those originating outside the mPFC. When dividing the overall projection density of mPFC areas (MO, PL/IL, AC) by non-mPFC areas (TeA, Rs) we obtained a ratio of 13.1, indicating that the mPFC projection to the DRN is 13.1 times higher than the non-mPFC one. We consider this likely indicates a stronger involvement of mPFC areas in the observations described in this study. However, we agree with the reviewer in this criticism and we have now provided a new version in which we limit our interpretation to cortico-DRN input, leaving the case of mPFC-DRN input as a likely scenario given our results, but not ruling out alternative explanations:

Lines 530-535: “Furthermore, we localized the origin of these projections and quantified the local neuronal loss. The PL, IL, and AC cortices, areas that comprise the so called mPFC (Klune et al., 2021), suffered a significant loss with the procedure. Although we cannot rule out the contribution of the other affected areas (Rs and TeA) in our caspase manipulation experiment, it is very likely that the areas with higher neuronal loss (mPFC) made a pivotal contribution to the behavioral changes we observed. ”

3. The description of the behavioral task is confusing and seems to obscure the fact that (based on the high probability of reward and high probability of switch used-0.9 each) the optimal strategy is to be flexible in response location, rather than persistent. More details are needed on the task: what determines trial start? What is the optimal strategy to use? What is the average number of responses before a subject should switch? Why do the authors state that this task "requires persistence in poking at the port despite reward failures" when it appears subjects are meant to be switching response locations frequently?

There are no signals to the mouse that a trial has started. The only feedback the mouse gets about the state of the port is the availability of water itself. In this sense is it not really a task with correct and incorrect trials signaled by clear feedback as in psychophysical tasks. We chose this deliberately because it resembles more closely a natural foraging situation and, perhaps for this reason, was quick for the mice to begin performing.

The reviewer is correct that the task calls for a relatively short amount of time poking compared to a version where the switch probability was lower. It is not simple to calculate the true optimal strategy in this task and we make no arguments concerning optimality. This is the case because the cost function of the mouse is not readily known. We do not claim or intend to imply that juvenile mice are less optimal or that the cortico-raphe pathway has an effect on task “success” or optimality. In fact, juveniles and adults receive a similar number of rewards per trial. We are solely using the task to measure persistence, which we define, following previous studies (Vertechi et al. Neuron 2020; Lottem et al., Nat. Comm. 2018), by the duration or number of pokes at a site. Persistence does not have an absolute reference point. A mouse may be slightly more persistent by showing a small increase in a relatively short time scale behavior, as the one we studied. At p = 0.9 switching probability, switching after 1 poke, i.e. simply alternating, would be ‘correct’ 90% of the time, which could be close enough to optimal to avoid a more complex strategy. Yet to avoid the cost of switching, an animal in a stochastic environment, which might also be changing over time (even though the actual environment we used was stationary) would require making more than 1 poke per side at least some of the time in order to collect information about the dynamics of the ports. This information seeking behavior is likely to be part of optimal behavior. Indeed, even juveniles make more than 1 poke per trial on average. We could speculate as to exactly what the raphe and cortico-raphe projection contribute to the orchestration of this behavior, but this is aside the point of our claims, which are solely about the tendency to stay and wait or repeat the same action rather than leaving. We now clarify this in the text:

Lines 238-259: “To investigate the development of behavioral persistence in mice, we employed a self-paced probabilistic foraging task (Vertechi et al., 2020). The setup consists of a box with two nose-ports separated by a barrier (Figure 2A). Each nose-port constitutes a foraging site that water-deprived mice can actively probe in order to receive water rewards. Only one foraging site is active at a time, delivering reward with a fixed probability. Each try in the active site can also cause a switch of the active site’s location with a fixed probability (Figure 2B). After a state switch, mice have to travel to the other port to obtain more reward, bearing a time cost to travel. In this task, a trial is defined as a bout of consecutive attempts on the same port, before leaving, and the amount of time spent attempting to obtain reward in one port before switching is the primary measure of persistence, independent of the specific strategy used by the mice (see Discussion).

We compared the behavior of juvenile (weeks 3-4) and adult mice (weeks 7-8) on their first exposure to the apparatus and task, ensuring that differences in persistence do not arise from differences in learning about the task. We used an environment characterized by high reward probability (Prwd = 90%) and high site-switching probability (Psw = 90%) (Figure 2A). These statistics produce a small number of rewards per trial (Rewards per trial: minimum = 0, maximum = 3) (Figure 2B), and maximize the number of trials performed in one session. Both groups obtained a comparable reward rate (Figure 2C; Adults: median = 0.02 rewards per second, 95% CI = [0.0020, 0.003], Juveniles: median = 0.023 rewards per second, 95% CI = [0.001, 0.012]; Mann-Whitney U test (N Adults = 21, N Juveniles = 23) = 182.0, p = 0.16), indicating that juveniles and adults do not differ in terms of overall competence in performing the task. ”

4. The text describes that each response in the active port carries a 0.9 probability that the active site will switch to the other location. This would seem to encourage alternation behavior after about 2 responses, not persistent behavior in one location. Are these data, therefore, reflecting maladaptive perseveration in adults, rather than adaptive persistence?

While it is difficult to quantify what is optimal behavior in this task and we did not attempt to do so, we fully agree with the reviewer that we cannot make conclusions about juveniles being ‘worse’ than adults in this foraging environment. It is important to note that both groups of mice are naive to the task and perform under a suboptimal “stimulus bound” strategy instead of an “inferencebased” one, as described by Vertechi et al. Neuron 2020. In the revised manuscript we clarify this in the manuscript:

Lines 486-498: “In line with pre-adolescent humans’ lack of delay gratification ability (Mischel et al., 1989), and with studies assessing impulsive behavior in mice over development (Sasamori et al., 2018), we found that mice of 3-4 weeks of age tend to be less persistent than 7-8 weeks old mice in a probabilistic foraging task. In a previous study we showed that adult mice are capable of performing the task by adopting an effective inference-based strategy (Vertechi et al., 2020) which involves tolerating a fixed number of consecutive failures after the last received reward independent of the total number of rewards obtained in that trial. This strategy is optimal because the state switch probability is independent of the reward probability. However, before learning this strategy, mice use a simpler “stimulus-bound” strategy in which the number of rewards received tends to increase persistence during a trial (Vertechi et al., 2020). Altogether, our observations suggest that naive juvenile and adult mice forage in a similar manner

5. There are key behavioral outcome measures missing from the analysis that would make it much stronger. Mean number of rewards earned per response should be provided to give the reader more information about how efficiently the juvenile vs. adult strategies performed.

We thank the reviewer for this point. We now include the suggested outcome measure, which shows that indeed the juvenile and adult mice have similar efficiency in terms of reward per trial. We also highlight this in the discussion as described above (See response to reviewer 1).

6. Some background on what is known about juvenile cognition would help put the results in context and appreciate any potential confounds. For example, juveniles are more impulsive.

We have now expanded the introduction of our manuscript to include several references (from rodents, primates and humans) illustrating the point raised by the reviewer:

Lines 52-60: “For instance, humans and macaques with prefrontal cortical damage display deficits in behavioral flexibility, decision making, and emotional processing (Izquierdo et al., 2017; Rudebeck et al., 2013; Roberts et al., 1998), as well as a notable increase in impulsive behavior (Berlin, 2004; Dalley and Robbins, 2017; Fellows, 2006; Itami and Uno, 2002), all of which, at least partially recapitulate features of juvenile behavior over healthy development in humans, non-human primates, and rodents (Rosati et al., 2023; Doremus-Fitzwater et al., 2012; Romer, 2010; Weed et al., 2008). In line with this, local pharmacological inhibition of mPFC significantly limits rats’ ability to wait for a delayed reward (Murakami et al., 2017; Narayanan et al., 2006).”

Lines 62-67: “Crucially, the mPFC undergoes intense postnatal maturation from childhood to adulthood, particularly during adolescence (Chini & Hanganu-Opatz., 2021), which in humans spans from years ~10-18 of life and in mice from weeks ~3-8 of life, and is a period of intense somatic maturation, including sexual development (Bell, 2018), and that correlates with a decrease in impulsive behavior characteristic of the juvenile phase (Rosati et al., 2023; Doremus-Fitzwater et al., 2012; Hammond et al., 2012; Konstantoudaki et al., 2018).”

7. In addition, the Vertechi et al., (2020) paper referenced for this behavioral task states that, "We found that the number of consecutive failures since the last reward (ConsecutiveFailureIndex) was a better predictor of mouse choice than the time spent at the nose poke". Number of consecutive failures should be reported here. Including this more informative measure is important especially when the time-based measures used here were vulnerable to task-irrelevant exploratory behavior, which skewed foraging episode times, and required the data the be reanalyzed with an arbitrary cutoff time of 60s.

In adult mice which are very familiar with the task, pokes rather than time is the best predictor of behavior. In untrained and juvenile mice we found that this was not the case. This might reflect the fact that early in task exposure, mice do not “realize” that individual pokes trigger water delivery and state changes. It is true that time-based measures are more susceptible to taskirrelevant behavior. We explored carefully various alternative measures and found generally consistent effects regardless of measure and precise cutoff values. As requested by the reviewer, we have now included the “pokes after last” measure (See below and Figures 2F and 3E of the updated manuscript) and excluded the analysis using an arbitrary cutoff. We hope the reviewer will agree that the consistency of results strengthens the robustness of our interpretation.

8. The measure of time spent poking does not seem informative, or at least it does not give the reader more information than the number of pokes produced in a given bout.

In adult mice which are very familiar with the task, pokes rather than time is the best predictor of behavior. In untrained and juvenile mice we found that this was not the case. This might reflect the fact that early in task exposure, mice do not “realize” that individual pokes trigger water delivery and state changes. It is true that time-based measures are more susceptible to taskirrelevant behavior. We explored carefully various alternative measures and found generally consistent effects regardless of measure and precise cutoff values. As requested by the reviewer, we have now included the “pokes after last” measure (See Author response image 1 and Figures 2F and 3E of the updated manuscript) and excluded the analysis using an arbitrary cutoff. We hope the reviewer will agree that the consistency of results strengthens the robustness of our interpretation.

9. The ablated pathway was not mPFC-DRN specific, but instead targeted any Rbp4-expressing neurons projecting to the DRN. The authors concluded that the highest density of DRN projections originated in mPFC, but only provided data from a small number of cortical comparison regions (temporal association cortex, retrosplenial cortex, and M1). Furthermore, somas expressing the control rAAV were shown to send collaterals to several regions outside of the DRN, such as the VTA, PAG, and the striatum. The loss of these collaterals in the ablation group could impact behavioral outcomes. Therefore, the data presented here are not sufficient to conclude that observed behavioral effects were specifically driven by mPFC-DRN pathway ablation. The language attributing behavioral changes exclusively to mPFC-DRN pathway ablation should be changed to reflect this lack of anatomical specificity.

Given that the Rbp4 line expresses Cre in all cortical areas, and being aware that any Rbp4+ neuron projecting to the DRN would be susceptible to be infected by rAAV vectors, we performed a careful characterization of the areas in which we found robust expression of tdTomato in control animals. In our hands, only 5 areas showed tdTomato+ neurons in at least 5/8 control mice, these areas were PR/IL, AC, MO, TeA and Rs. Areas showing expression in 1-3 mice were not included for analysis. This information has been included in the methods and results for clarity:

Methods:

Lines 696-701: “We found five cortical areas consistently expressing layer 5 tdTomato+ neurons in at least 5/8 control mice: PR/IL, AC, MO, TeA, Rs. We then acquired confocal images of these 5 areas in mice injected with rAAV-tdTomato or rAAV-Caspase for analysis. Areas showing expression in 1-3 mice were not included for analysis”

Results:

Lines 364-391: “We found tdTomato+ expressing neurons in five cortical areas (Figure S4AG), which is largely consistent with previous reports (Xu et al., 2021). Furthermore, these neurons mainly originated in the prefrontal cortex, being 13 times more abundant than those from other cortical areas outside the prefrontal cortex. The prelimbic/infralimbic (PL/IL) and anterior cingulate (AC) cortices, which constitute the mPFC, were the areas with the highest density of DRN-projecting tdTomato+ somas in control animals (Figure S4B-D-E, median = 3.41 neurons per layer 5 bin, 95% CI = [1.45, 3.81] for PL/IL and median = 1.54 neurons per layer 5 bin, 95% CI = [1.18, 3.64] for AC) and consistently more extensive neuron density loss in caspase injected mice, quantified using the pan-neuronal marker NeuN (Figure 3A-C, control n=8 mice vs. caspase n=7 mice, two-sample Kolmogorov-Smirnoff Test = 0.028, p = 0.002 for PL/IL and D = 0.024, p = 0.01 for AC). We also found tdTomato+ somata in the medial orbitofrontal cortex (MO) of the control group; however, this projection was less robust in terms of tdTomato+ labeled neurons across animals (Figure S4B-C, median = 1.34 neurons per layer 5 bin, 95% CI = [0.37, 4.81]) and, consistently, the difference in layer 5 NeuN densities between control and caspase mice was not significant (Figures 3C, S4, D = 0.017, p = 0.08).

Apart from the mPFC, sparse labeling of tdTom+ neurons was found in more posterior levels of the neocortex, namely in the retrosplenial cortex (RS) and in the temporal association cortex (TeA) (Figure S4B,F,G; median = 0.11 neurons per layer 5 bin, 95% CI = [0.0, 0.56] for Rs and median = 0.0 neurons per layer 5 bin, 95% CI = [0.0, 0.52] for TeA). However, it is worth noting that tdTom+ neurons were only found in the RS of 5 out of 8 control animals, and in the TeA of 3 out of 8 control animals. Consistently, the reduction in NeuN layer 5 neuronal density in these two areas was minimal and non-significant compared to controls (Figure 3C, D = 0.034, p = 0.12 for RS and D = 0.025, p = 0.19 for TeA). In addition, no differences in NeuN density were observed between caspase- and tdTomato-injected animals in an area that does not contain tdTomato expressing somas and therefore not projecting to the DRN which serves as a negative control to rule out unspecific biases in our quantification method (M1, Figure 3C, D = 0.019, p = 0.15). These observations suggest that our ablation approach primarily affected mPFC-DRN projecting neurons, particularly from PL/IL and AC cortices.”

M1 was included as a negative control for the specificity in the method of quantifying NeuN densities. Given that M1 did not contain tdTomato+ neurons in any control animal, we should expect that M1 is not susceptible to present layer 5 cell death in Caspase injected mice compared to controls. As expected, we observed comparable NeuN density in control and caspase injected mice in M1, which indicated that our NeuN quantification method did not show any systematic artifactual bias:

Lines 386-391: “In addition, no differences in NeuN density were observed between caspase- and tdTomato-injected animals in an area that does not contain tdTomato expressing somas and therefore not projecting to the DRN which serves as a negative control to rule out unspecific biases in our quantification method (M1, Figure 3C, D = 0.019, p = 0.15). These observations suggest that our ablation approach primarily affected mPFC-DRN projecting neurons, particularly from the PL/IL and AC cortices.”

Regarding the characterization of collateral projections affected in the caspase manipulation, we note that this is an important and often overlooked piece of information in many studies using “pathway specific” optogenetic stimulation of axons without comment on the possibility that this activates collaterals through action potential backpropagation. We completely agree with the reviewer that we cannot unequivocally attribute the phenotypic effect of the ablation to only one pathway (as we emphasize in the discussion), but in comparison to the standards for studies using similar methods to address similar questions we would suggest that ours is not less precise than many others making as or more specific claims.

10. The quantification of NeuN-expressing cell density was not described in the methods or main text. Since NeuN-labeling was used to quantify/confirm neuronal density loss in ablation mice compared to controls, the quantification process should be described fully in the methods.

We thank the reviewer for pointing out this issue and we apologize for not having included this important information in the submitted manuscript. We have now incorporated this information into the methods section.

Lines 696-706: “Quantification of NeuN expressing neurons was performed using the same protocol used in Rbp4-tdTomato mice. First, we visually inspected the expression pattern of tdTomato expressing neurons after injecting rAAV-tdTomato in the DRN of control mice. We found 5 cortical areas consistently expressing layer 5 tdTomato+ neurons in at least 5/8 control mice: PR/IL, AC, MO, TeA, Rs. We then acquired confocal images of these 5 areas in mice injected with rAAV-tdTomato or rAAV-Caspase for analysis. Areas showing expression in 1-3 mice were not included for analysis. These confocal stacks contained the green fluorescent signal of NeuN detection and the red intrinsic fluorescence of tdTomato and were all constant in size. Using custom made software based on Matlab's image analysis toolbox, NeuN somas were detected and their densities binned in depth and averaged across mice for final representation. ”

11. The kernel density functions for 'leaving time' appear bimodal in several cases, especially in Figure 2C. The possibility that individuals within the same group/condition are using different strategies or are displaying different behavioral pattens should be addressed.

We thank the reviewer for pointing this out. As noticed by the reviewer, the mean group distributions of leaving times exhibit a bimodal pattern with long and short leaving durations (Figure S3A) which could potentially stem from multiple sources.

**Author response image 2. sa2fig2:** Kernel density function of leaving times of control animals, in blue, and experimental animals, in red. Solid and dashed lines indicate animal and group functions, respectively.

One possibility is that animals, being new to the box environment, intermingle trials of focused foraging with those involving exploratory interactions with the box. Alternatively, it could arise from averaging individual mice adopting distinct strategies. To investigate the homogeneity within groups, i.e., to validate if individual animals within a group exhibit both strategies, we employed k-means clustering to categorize trials into long and short leaving times at the group level:

**Author response image 3. sa2fig3:** K-means based categorization of each trial in short and long leaving times. Individual control and experimental animals are colored with shades of blue and red respectively.

Next we employed Fisher's exact test to evaluate the null hypothesis that the likelihood of individual mice displaying long leaving times is equivalent to the remainder of their respective groups. Applying this methodology, we identified that only one adult animal (Plong-leaving = 0.37, p = 0.0372) and one caspase animal (Plong-leaving = 0.29, p = 0.0016) exhibited a significantly lower probability of adopting extended leaving times in comparison to their respective groups. Subsequently, we re-executed the leaving time logistic regression analysis in the main text after excluding these two animals. Notably, the exclusion did not induce any alteration in the identification of significant factors (data not included). Overall, these findings indicate that even though differences between short and long leaving times exist (as expected given their phenotype), all animals engage in both types of trials (see Supplementary Figure 3A). This contributes to a cohesive behavioral pattern within each group. Furthermore, our analytical approach to estimate the likelihood of leaving following each poking action remains robust against these features of our dataset.

**Author response image 4. sa2fig4:** Scatter plot of the difference in the frequency use of short and long leaving time for each animal. Control and experimental animals are colored in blue and red respectively.

Lines 1243-1250: “Supplementary Figure 3. Description of poking behavior over the session progression and according to sex. (A) Distribution of the trial durations for naive juveniles and naive adults (left) and for Caspase and tdTomato control mice (right). Note that the bimodality of the data visibly arises at the single mouse level, indicating each mouse performs short and long leaving times. Indeed, when using Fisher's exact test to evaluate the null hypothesis that the likelihood of individual mice displaying long leaving times (obtained using k=2 K-means based categorization) is equivalent to the remainder of their respective groups, 57/59 mice were unable to reject the null hypothesis (data not shown).”

[Editors’ note: what follows is the authors’ response to the second round of review.]

Comments to the Authors:While we appreciated your responses to the original reviews, because of the very long time between the initial submission and this revision, some of the prior reviewers were not available. As a result, it was necessary to obtain new reviewers, who raised additional substantial concerns regarding some of the methodology. They felt that these issues required substantial additional control work.Reviewer #2 (Recommendations for the authors):This study characterizes the development of cortico-raphe projections from L5 cortical neurons and their influence on foraging and persistent behavior. The finding that cortico-raphe projections mature late in development is extremely interesting and raises several new questions regarding the role of these projections in adolescence. Most of the significant points have already been addressed by other reviewers; I will focus on a few technical aspects of the electrophysiology experiments and the caspase3-based approach.The late development or cortico-raphe projections is well supported by the histological and optogenetic experiments depicted in Figure 1. However, the reported oEPSC values seem to be based on a single trial/stimulus, is that correct? The authors mention "TTL-triggered pulses of light (10-ms duration; 10 mW measured at the fiber tip) were delivered at the recording site with a 10-second inter-sweep interval. In cases where multiple pulses were delivered per sweep, only the first one was considered for analysis to eliminate short-term plasticity effects on the measured amplitude." – A 10-second inter-sweep interval is insufficient to recover opsin desensitization, potentially leading to the perceived short-term plasticity. Increasing the inter-sweep interval to >45 seconds should resolve this issue and enable the averaging of individual sweeps to minimize the variability inherent in opto-evoked whole-cell recordings.

We thank the reviewer for pointing this out and we apologize if the phrasing used in the optogenetic circuit mapping methods led to confusion. Every data point represented in figure 1B-C corresponds to the average of 6-10 sweeps (trials) per recording (with an intersweep interval of 10 seconds). The phrase “*In cases where multiple pulses were delivered per sweep, only the first one was considered for analysis to eliminate short-term plasticity effects on the measured amplitude*” present in the methods refers to a subset of recorded neurons in which instead of a single optogenetic pulse per sweep, the stimulus consisted of a train of light pulses delivered at frequencies ranging from 2-10Hz. In this subset of recordings (<10% of the total of recorded neurons), only the amplitude in response to the first peak of the stimulus train was considered for amplitude analysis.

To address the reviewer's concern about possible intersweep desensitization issues of the opsin in our recording conditions, we have plotted the amplitudes per consecutive sweep normalized to the amplitude of the first sweep per recording for the 63 connected neurons reported in this manuscript throughout all ages. As it can be seen, no sign of desensitization (decrease of oEPSC amplitude with consecutive sweeps) can be observed (see Author response image, panel A). Furthermore, this holds true for the neurons in which a train of light pulses was delivered (see Author response image, panel A and B). The neuron depicted in panel B, despite its low connection amplitude does not show any sign of desensitization over consecutive sweeps (data represented in blue line is the amplitude per sweep of the first oEPSC of the stimulus train normalized to the first sweep). These recordings represent 9/63 of the reported connected neurons (all ages), the remaining 54 had a single pulse as illustrated in Author response image 5, panel C.

**Author response image 5. sa2fig5:** 

To clarify this point, we added the following explanation to the methods section:Lines 578-587: “TTL triggered pulses of light (10-ms duration; 10 mW measured at the fiber tip) were delivered at the recording site with 10 seconds of intersweep interval. In >90% of the neurons considered in the current study, the stimulus consisted in a single pulse of light per sweep. In the remaining subset of recorded neurons the stimulus consisted of a train of light pulses, of same length and amplitude, delivered at frequencies ranging from 2-10Hz. In this subset of recordings, only the amplitude in response to the first peak of the stimulus train was considered for amplitude analysis. Importantly, no sign of intersweep opsin desensitization (decrease of light evoked EPSC amplitude with consecutive sweeps) was observed in either type of recordings (data not shown). Every data point represented in figure 1B-C corresponds to the average of 6-10 sweeps per recording”.

The control experiments based on NeuN IHC do not provide an estimate of the fraction of cortico-raphe projections ablated by the retro caspase approach, which is an important aspect to better interpret the behavioral results. This is because AAVrg only infects a subset of neurons. There are three potential alternative approaches to characterize the extent of cortico-raphe projection loss. One is to perform a second AAVrg-tdTom injection two weeks after AAVrg-DIO-casp3 to compare how many cells can still be retro-infected after the first caspase injection. However, quantifying these experiments is not straightforward, and variability due to viral injection efficiency and targeting is always a confounding factor. A more rigorous and cleaner approach could be to inject AAVrg-DIO-casp3 in Rbp4-Cre x Ai32 mice and compare oEPSC amplitudes in injected vs non-injected mice. As ChR2 is stably expressed in these mice this would provide a better assessment of connectivity changes after caspase ablations. A third approach would be to use Rbp4-Cre x XFP mice and quantify cortico-raphe fiber density changes after AAVrg-DIO-casp3 (similar to what is shown in Figure 1).However, these strategies would only probe the fraction of cortical neurons expressing Cre, which, in the case of Rbp4-Cre, is approximately 50% of L5 neurons. It's possible that Rbp4-negative L5 neurons also project to the raphe, and these are not affected by DIO-casp3 ablation. Quantifying the extent of total cortico-raphe ablation is an important point, as considerable remaining fibers could lead to an underestimation of the behavioral effects caused by cortico-raphe manipulations.

In response to points 2 and 3, Tervo and colleagues (Neuron, 2016) reported that a Cre-dependent fluorescent reporter expressing retroAAV injected in the basal pontine nuclei of Rbp4-Cre mice produces a comparable density of labeled layer 5 cortical neurons as obtained with a standard retrograde tracer such as fluorogold. This suggests that the Rbp4 promoter grants genetic access to virtually all layer 5 projecting neurons, at least in the case of cortico-pontine projection neurons. However, we cannot conclude that this holds true for the case of cortico-raphe projections. Therefore, in response to the reviewer’s concern, we have now added the following sentence to our manuscript to clearly state this possibility:

Lines 454-465: “It should be noted that the extent of layer 5 neurons affected by the caspase ablation in these cortical areas will be defined by the total percentage of layer 5 neurons expressing Rbp4. A previous study has shown that a Cre-dependent fluorescent reporter expressing retroAAV injected in the basal pontine nuclei of Rbp4-Cre mice produces a comparable density of labeled layer 5 cortical neurons as obtained with a standard retrograde tracer such as fluorogold (Tervo et al., 2016). This suggests that, at least for the case of cortico-pontine projection neurons, the Rbp4 promoter grants genetic access to virtually all layer 5 projecting neurons. However, we cannot conclude that this holds true for the case of cortico-raphe projections and therefore future work will have to assess whether additional non-Rbp4 populations of projecting neurons in these, or other cortical areas, contribute as well to the development of behavioral persistence”.

Unless I missed it, there is no description of the age at which the caspase injection is performed in the different experiments. This is an important experimental detail.

We thank the reviewer for pointing this out. We have now updated our methods section to include this information:

Lines 646-647: “Adult mice between 8 and 9 weeks of age were anesthetized with isoflurane (2% induction and 0.5 – 1% for maintenance) and placed in a motorized computer-controlled Stoelting stereotaxic instrument”.

Regarding the specificity of mPFC>raphe projections raised by other reviewers, the authors discuss the possibility of other mPFC collaterals in the striatum. Future experiments inhibiting local terminals (perhaps using Emx1-Cre crosses) with PPO/eOPN3 or activating ChR2 fibers in juveniles will help further support these findings.

While we agree with the reviewer on this point, as openly discussed in our manuscript, the fact that cortico-striatal connections are fully developed by p14 (Peixoto et al., Nat. Neurosci. 2016) makes it unlikely that these projections per se are responsible for the persistent behavior observed in adults. We believe that the fact that juveniles have cortico-striatal mature synapses but not cortico-raphe mature synapses early during adolescence, strengthens the case for the role of developing cortico-raphe projections as a crucial process for the development of behavioral persistence. Thus, while we see the value of the experiments proposed by the reviewer, we believe they represent a large amount of work for what exceeds the scope of our results.

Reviewer #3 (Recommendations for the authors):The paper in general gives very little detail on the anatomical analysis,For the developmental description, It would be good to show to what extent the Rbp4 labels the PFC-raphe projection. Convincing would be for instance a retrograde labeling from the raphe in the Rbp4-GFP with some quantitative estimate.

In line with the concern expressed by reviewer 2, we have now included this text in the manuscript aiming to acknowledge the possibility that the Rbp4 promoter does not grant access to virtually all cortico-raphe projecting neurons:

Lines 454-465: “It should be noted that the extent of layer 5 neurons affected by the caspase ablation in these cortical areas will be defined by the total percentage of layer 5 neurons expressing Rbp4. A previous study has shown that a Cre-dependent fluorescent reporter expressing retroAAV injected in the basal pontine nuclei of Rbp4-Cre mice produces a comparable density of labeled layer 5 cortical neurons as obtained with a standard retrograde tracer such as fluorogold (Tervo et al., 2016). This suggests that, at least for the case of cortico-pontine projection neurons, the Rbp4 promoter grants genetic access to virtually all layer 5 projecting neurons. However, we cannot conclude that this holds true for the case of cortico-raphe projections and therefore future work will have to assess whether additional non-Rbp4 populations of projecting neurons in these, or other cortical areas, contribute as well to the development of behavioral persistence”.

For the analysis of afferent axons, the methods authors state that the analysis was done on sagittal sections, in which the position of the DRN and accumbens is not very easy to identify. Yet In figure 1 they show coronal sections.

We thank the reviewer for pointing out this mistake in the methods section. No sagittal slices were obtained in the present study (only coronal, as shown in figure 1). We have now corrected this in the methods.

Lines 598-599: “Coronal sections (50 µm) were cut with a freezing sliding microtome (SM2000, Leica)”.

Additionally, at this resolution and without co-labeling with a synaptic marker they cannot distinguish an increase of fluorescence or passing fibres from a true increase in the number of terminals.

While we agree with the reviewer’s point that axonal densities do not necessarily reflect the density of synaptic contacts (given the lack of specificity between passing vs. connecting axons), we believe that our electrophysiology results using optogenetic circuit mapping addresses this point more precisely than using immunohistochemistry for synaptic markers. The fact that the axonal densities reported and optogenetic circuit mapping responses show a consistent progression over adolescence, strengthens our point about a developing cortico-raphe pathway.

For the retrograde lesion studies, they need to show the injection site of their viral delivery to determine how this could have impacted neighbouring structures. They also could better quantify the specific loss of l5 neurons with an independent L5 marker such as Ctip2.

In line with the reviewer’s suggestion, we have now included injection site examples and a quantification of NeuN density at the site of injection (Figure 3 Suppl. 1). Since both viruses are retrograde and Cre dependent, the injection site usually contains none, or just a handful of locally transfected neurons (see Figure 3 Suppl. 1). Consistent with this, the density of NeuN neurons in the injection site is comparable across groups (see Figure 3 Suppl. 1, Caspase average binned NeuN densities subtracted to tdTomato controls show values close to zero), indicating absence of neuronal loss. Moreover, in all control injections we found a dense cortical axonal innervation specifically over the DRN, which is consistent with cortico-DRN projections being retrogradely labeled (see below binned fluorescent signal from cortical axons over the DRN).

In all the morphological analysis they need to show where the measures were done. They also need to indicate more precisely how the measures were made (field size analyzed, precise steps of image processing, magnification, number of sections analysed /case).

We thank the reviewer for pointing this out. We have now updated our methods section to include this information.

Lines 635-641: “We found five cortical areas consistently expressing layer 5 tdTomato+ neurons in at least 5/8 control mice: PR/IL, AC, MO, TeA, Rs. We then acquired confocal images of these 5 areas in mice injected with rAAV-tdTomato or rAAV-Caspase for analysis. Areas showing expression in 1-3 mice were not included for analysis. These confocal stacks contained the green fluorescent signal of NeuN detection and the red intrinsic fluorescence of tdTomato. All confocal images consisted of 10 images stacked in the Z plane, with 3 μm spacing, and that were max projected for analysis. Stacks from PR/IL, AC, MO were 800x600 μm (cortical depth x width) and from TeA, Rs and M1 were 1400x600 μm to adjust to their intrinsically different cortical thickness (Figure 3 Suppl. 2). For each brain area/mouse, bilateral stacks were acquired at Bregma levels: PR/IL: 1.5 mm, AC and M1: 1.1 mm, MO: 2.3 mm, TeA and Rs: -3.1 mm. Using custom made software based on Matlab's image analysis toolbox, NeuN somas were detected and their densities binned in depth and averaged across mice for final representation”.